# EFFICIENT APPROXIMATIONS OF COMPLETE INTERATOMIC POTENTIALS FOR CRYSTAL PROPERTY PREDICTION

## ABSTRACT

We study the problem of crystal material property prediction. A crystal structure consists of a minimal unit cell that is repeated infinitely in 3D space. How to accurately represent such repetitive structures in machine learning models remains unresolved. Current methods construct graphs by establishing edges only between nearby nodes, thereby failing to faithfully capture infinite repeating patterns and distant interatomic interactions. In this work, we propose several innovations to overcome these limitations. First, we propose to model physics-principled interatomic potentials directly instead of only using distances as in existing methods. These potentials include the Coulomb potential, London dispersion potential, and Pauli repulsion potential. Second, we propose to model the complete set of potentials among all atoms, instead of only between nearby atoms as in prior methods. This is enabled by our approximations of infinite potential summations with provable error bounds. We further develop efficient algorithms to compute the approximations. Finally, we propose to incorporate our computations of complete interatomic potentials into message passing neural networks for representation learning. We perform experiments on the JARVIS and Materials Project benchmarks for evaluation. Results show that the use of complete interatomic potentials leads to consistent performance improvements with reasonable computational costs.

## 1 INTRODUCTION

The past decade has witnessed a surge of interests and rapid developments in machine learning for molecular analysis (Duvenaud et al., 2015). These initial studies mainly focus on the prediction and generation problems of small molecules. To enable computational analyses, molecules need to be featurized in an appropriate mathematical representation form. Recently, with the advances of graph neural networks (GNNs) (Gilmer et al., 2017; Battaglia et al., 2018), molecules are more commonly represented as graphs in which each node corresponds to an atom, and each edge corresponds to a chemical bond (Stokes et al., 2020; Wang et al., 2022c). A variety of molecular graph prediction (Stokes et al., 2020; Wang et al., 2022c) and generation (Shi et al., 2019; Jin et al., 2018; Luo et al., 2021) methods have been developed based on 2D molecular graph representations. A key limitation of the 2D graph representations is that the 3D geometries of molecules are not captured, but such information may be critical in many molecular property prediction problems (Hu et al., 2021). To enable the encoding of 3D molecular geometries in GNNs, a series of 3D GNN methods have been developed for prediction (Schütt et al., 2017; Gasteiger et al., 2019; Liu et al., 2022b; Wang et al., 2022b) and generation (Liu et al., 2022a; Luo & Ji, 2022; Hoogeboom et al., 2022) problems. In these 3D graph representations, each node is associated with the corresponding atom's coordinate in 3D space. Geometric information, such as distances between nodes and angles between edges, is used during message passing in GNNs. Recently, these methods have been extended to learn representations for proteins (Jing et al., 2020; Wang et al., 2022a).

Inspired by the success of GNNs on small molecules, Xie & Grossman (2018) developed the crystal graph convolutional neural network (CGCNN) for crystal material property prediction. Different from small molecules and proteins, crystal materials are typically modeled by a minimal unit cell (similar to a small molecule) that is repeated in 3D space with certain directions and step sizes. In theory, the unit cell is repeated infinitely in 3D space, but any real-world material has finite size.

However, given that our modeling is at the atomic level, modeling crystal materials as infinite repetitions of unit cells is approximately accurate. Therefore, a key challenge in crystal material modeling is how to accurately capture the infinite-range interatomic interactions resulted from the repetitions of unit cells in 3D space. Current GNN-based crystal property prediction methods construct graphs by creating edges only between atoms within a pre-specified distance threshold (Xie & Grossman, 2018; Chen et al., 2019; Louis et al., 2020; Schmidt et al., 2021; Choudhary & DeCost, 2021). Thus, they fail to capture interactions between distant atoms explicitly.

In this work, we propose a new graph deep learning method, **PotNet**, with several innovations to significantly advance the field of crystal material modeling. First, we propose to model interatomic potentials directly as edge features in PotNet, instead of using distance as in prior methods. These potentials include the Coulomb potential (West, 1988), London dispersion potential (Wagner & Schreiner, 2015), and Pauli repulsion potential (Krane, 1991). Second, a distinguishing feature of PotNet is to model the **complete** set of potentials among all atoms, instead of only between nearby atoms as in prior methods. This is enabled by our approximations of infinite potential summations with provable error bounds. We further develop efficient algorithms to compute the approximations. Finally, we propose to incorporate our computations of complete interatomic potentials into message passing neural networks for representation learning. We performed comprehensive experiments on the JARVIS and Materials Project benchmarks to evaluate our methods. Results show that the use of complete interatomic potentials in our methods leads to consistent performance improvements with reasonable computational costs.

## 2 BACKGROUND AND RELATED WORK

### 2.1 CRYSTAL REPRESENTATION AND PROPERTY PREDICTION

A crystal structure can be represented as periodic repetitions of unit cells in the three-dimensional (3D) Euclidean space, where the unit cell contains the smallest repeatable structure of a given crystal. Specifically, let $n$ be the number of atoms in the unit cell, a crystal can be represented as $\boldsymbol{M} = (\boldsymbol{A}, \boldsymbol{L})$. Here, $\boldsymbol{A} = \{\boldsymbol{a}_i\}_{i=1}^n = \{(\boldsymbol{x}_i, \boldsymbol{p}_i)\}_{i=1}^n$ describes one of the unit cell structures of $\boldsymbol{M}$, where $\boldsymbol{x}_i \in \mathbb{R}^b$ and $\boldsymbol{p}_i \in \mathbb{R}^3$ denote the $b$-dimensional feature vector and the 3D Cartesian coordinates of the $i$-th atom in the unit cell, respectively. $\boldsymbol{L} = [\boldsymbol{l}_1, \boldsymbol{l}_2, \boldsymbol{l}_3] \in \mathbb{R}^{3 \times 3}$ is the lattice matrix describing how a unit cell repeats itself in the 3D space. In the complete crystal structure, every atom in a unit cell repeats itself periodically in the 3D space. Specifically, from an arbitrary integer vector $\boldsymbol{k} \in \mathbb{Z}^3$ and the unit cell structure $\boldsymbol{A}$, we can always obtain another repeated unit cell structure $\boldsymbol{A}^{\boldsymbol{k}} = \{\boldsymbol{a}_i^{\boldsymbol{k}}\}_{i=1}^n = \{(\boldsymbol{x}_i^{\boldsymbol{k}}, \boldsymbol{p}_i^{\boldsymbol{k}})\}_{i=1}^n$, where $\boldsymbol{x}_i^{\boldsymbol{k}} = \boldsymbol{x}_i$, $\boldsymbol{p}_i^{\boldsymbol{k}} = \boldsymbol{p}_i + \boldsymbol{L}\boldsymbol{k}$. Hence, the complete crystal structure $\widetilde{\boldsymbol{A}}$ of $\boldsymbol{M}$ with all unit cells can be described as

$$\widetilde{\boldsymbol{A}} = \bigcup_{\boldsymbol{k} \in \mathbb{Z}^3} \boldsymbol{A}^{\boldsymbol{k}}. \tag{1}$$

In this work, we study the problem of crystal property prediction. Our objective is to learn a property prediction model $f : \boldsymbol{M} \to y \in \mathbb{R}$ that can predict the property $y$ of the given crystal structure $\boldsymbol{M}$. We will focus on predicting the total energy, or other energy-related properties of crystals.

### 2.2 CRYSTAL PROPERTY PREDICTION WITH INTERATOMIC POTENTIALS

Most of the classical crystal energy prediction methods are based on interatomic potentials. According to the studies in physics (West, 1988; Daw et al., 1993; Brown, 2016), the total energy of a crystal structure can be approximated by the summation of interatomic potentials in the crystal. Particularly, the three following categories of interatomic potentials are widely used in crystals, and they are considered as sufficient for accurate energy approximation.

- **Coulomb potential** is caused by the electrostatic interaction of two atoms with charges. Coulomb potentials are closely related to ionic bonding and metallic bonding in crystals (West, 1988). For any two atoms $\boldsymbol{a}$ and $\boldsymbol{b}$, let $z_{\boldsymbol{a}}$ and $z_{\boldsymbol{b}}$ denote the number of charges in the atom $\boldsymbol{a}$ and $\boldsymbol{b}$, and let $d(\boldsymbol{a}, \boldsymbol{b})$ be the Euclidean distance between the atom $\boldsymbol{a}$ and $\boldsymbol{b}$. The Coulomb potential $V_{\text{Coulomb}}$ is defined as $V_{\text{Coulomb}}(\boldsymbol{a}, \boldsymbol{b}) = -\frac{z_{\boldsymbol{a}} z_{\boldsymbol{b}} e^2}{4\pi \epsilon_0 d(\boldsymbol{a}, \boldsymbol{b})}$. Here $e$ is the elementary charge constant, and $\epsilon_0$ is the permittivity constant of free space.

- **London dispersion potential** describes the Van der Waals interaction between atoms. London dispersion potential is often considered in energy estimation because its contribution is cumulative over the volume of crystals (Wagner & Schreiner, 2015) and sometimes very strong in bulk crystals, such as the crystals containing sulfur and phosphorus. The mathematical form of this potential can be described as $V_{\text{London}}(\boldsymbol{a}, \boldsymbol{b}) = -\epsilon/d^6(\boldsymbol{a}, \boldsymbol{b})$, where $\epsilon$ is a hyperparameter.

- **Pauli repulsion potential** is resulted from the Pauli exclusion principle generally exists in all crystal structures. The Pauli exclusion principle forces any two atoms to be sufficiently far away from each other so that the electron orbits of them do not overlap. Such exclusion interactions lead to Pauli repulsion potential with the form of $V_{\text{Pauli}}(\boldsymbol{a}, \boldsymbol{b}) = e^{-\alpha d(\boldsymbol{a}, \boldsymbol{b})}$, where $\alpha$ is a hyperparameter (Buckingham, 1938; Slater, 1928).

### 2.3 CRYSTAL PROPERTY PREDICTION WITH DEEP LEARNING

While physics-based methods have been used for predicting the crystal energy for a long time, these methods are usually crystal-specific, i.e., one method can only achieve accurate approximation for one specific type of crystals. Recently, thanks to the advances of deep learning, many studies have been done to develop a general crystal property predictor for a variety of different crystals with powerful deep neural network models. Some studies (Wang et al., 2021; Jha et al., 2018; 2019; Goodall & Lee, 2020) represent crystals as chemical formulas, and adopt sequence models to predict properties from these string representations. However, more recent studies consider crystals as 3D graphs and employ expressive 3D GNNs (Schütt et al., 2017; Klicpera et al., 2020b; Liu et al., 2022b), a family of deep neural networks specifically designed for 3D graph-structured data, to crystal representation learning. CGCNN (Xie & Grossman, 2018) is the first method that proposes to represent crystals with radius graphs and adopts a graph convolutional network to predict the property from the graph. Based on the pioneering exploration of CGCNN, a lot of subsequent studies (Schmidt et al., 2021; Louis et al., 2020; Chen et al., 2019; Choudhary & DeCost, 2021; Batzner et al., 2022) propose various 3D GNN architectures to achieve more effective crystal representation learning. Particularly, by enhancing the input features with angle information, ALIGNN (Choudhary & DeCost, 2021) develops the currently most powerful 3D GNN architecture for crystals and achieves the best crystal property prediction performance.

## 3 METHOD

Although existing GNN-based methods have achieved impressive performance in crystal property prediction, they struggle in further boosting the performance due to the approximation of interatomic interactions using functional expansions based on distances and failing in capturing complete interatomic interactions. In this section, we present PotNet, a novel crystal representation model that can overcome these limitations of prior methods. Based on the physical modeling of crystal energy, PotNet explicitly uses infinite potential summations as input features to capture complete interatomic interactions. The infinite potential summations are incorporated into the message passing mechanism of graph neural networks and efficiently approximated by a fast algorithm. To the best of our knowledge, PotNet is the first work that bridges the classical crystal energy computation methods based on potentials and the data-driven methods based on deep neural networks.

### 3.1 APPROXIMATING CRYSTAL ENERGY WITH COMPLETE INTERATOMIC POTENTIALS

According to the density functional theory (DFT) in physics, for any crystal $\boldsymbol{M} = (\boldsymbol{A}, \boldsymbol{L})$ with the complete structure $\widetilde{\boldsymbol{A}}$ defined in Eqn. (1), its total energy $E(\boldsymbol{M})$ can be accurately approximated by the embedded atom method (Daw & Baskes, 1984; Daw et al., 1993; Baskes, 1987; Lee et al., 2016; Riffe et al., 2018) in the form of

$$E(\boldsymbol{M}) = \frac{1}{2} \sum_{\boldsymbol{a} \in \boldsymbol{A}} \sum_{\boldsymbol{b} \neq \boldsymbol{a}, \boldsymbol{b} \in \widetilde{\boldsymbol{A}}} V(\boldsymbol{a}, \boldsymbol{b}) + \sum_{\boldsymbol{a} \in \boldsymbol{A}} F(\rho_{\boldsymbol{a}}), \tag{2}$$

where $V(\boldsymbol{a}, \boldsymbol{b})$ denotes the interatomic potentials between the atoms $\boldsymbol{a}$ and $\boldsymbol{b}$, capturing the magnitude of interactions; $\rho_{\boldsymbol{a}}$ is the local electron density of the atom $\boldsymbol{a}$, determined by the coordinate and number of charges of the atom $\boldsymbol{a}$ according to the Hohenberg-Kohn theorem; $F(\cdot)$ is a parametrized

function to embed the electron density $\rho_a$. Actually, existing studies (Jalkanen & Müser, 2015) show that $\rho_a$ can be considered as a function of $\sum_{b\neq a, b\in\widetilde{A}} V(a, b)$ mathematically. Hence, Eqn. (2) can be rewritten in the following form:

$$E(\boldsymbol{M}) = \sum_{\boldsymbol{a}\in\boldsymbol{A}} \left[ \frac{1}{2} \sum_{\boldsymbol{b}\neq\boldsymbol{a}, \boldsymbol{b}\in\widetilde{\boldsymbol{A}}} V(\boldsymbol{a}, \boldsymbol{b}) + G\left( \sum_{\boldsymbol{b}\neq\boldsymbol{a}, \boldsymbol{b}\in\widetilde{\boldsymbol{A}}} V(\boldsymbol{a}, \boldsymbol{b}) \right) \right], \tag{3}$$

where $G(\cdot)$ is a parametrized function. Eqn. (3) can be considered as a way to compute the energy from the complete interatomic potential summation $\sum_{b\neq a, b\in\widetilde{A}} V(a, b)$ of every atom $a$ in the unit cell $A$. However, in practice, the function $G$ is computationally expensive if not infeasible. Hence, more and more studies have turned to the powerful learning capability of modern deep neural network models to approximate it effectively.

### 3.2 LIMITATIONS OF EXISTING DEEP LEARNING METHODS

Currently, most of the existing graph deep learning methods for crystals (Xie & Grossman, 2018; Chen et al., 2019; Louis et al., 2020; Choudhary & DeCost, 2021) use radius graph representations and distance-based features as inputs to predict the crystal energy in Eqn. (3). Specifically, for a crystal $\boldsymbol{M} = (\boldsymbol{A}, \boldsymbol{L})$, the radius graph is constructed by adding edges between any atom $\boldsymbol{a}$ in the unit cell $\boldsymbol{A}$ and any other atom $\boldsymbol{b}$ in the complete crystal structure $\widetilde{\boldsymbol{A}}$ whose distances are smaller than a pre-specified distance threshold $r$. In addition, some functional expansions of distances, e.g., radial basis functions (RBF), are used to model interatomic interactions and form the input edge features to 3D GNN models. Hence, let $\boldsymbol{a} = (\boldsymbol{x_a}, \boldsymbol{p_a}), \boldsymbol{b} = (\boldsymbol{x_b}, \boldsymbol{p_b})$, the crystal energy prediction $\hat{E}(\boldsymbol{M})$ of these methods can be generally described as

$$\hat{E}(\boldsymbol{M}) = \sum_{\boldsymbol{a}\in\boldsymbol{A}} \sum_{\boldsymbol{b}\in\mathcal{N}_r(\boldsymbol{a})} H\left(\phi\left(||\boldsymbol{p_a} - \boldsymbol{p_b}||_2\right)\right), \tag{4}$$

where $\mathcal{N}_r(\boldsymbol{a}) = \{\boldsymbol{b} : \boldsymbol{b} \neq \boldsymbol{a}, \boldsymbol{b} \in \widetilde{\boldsymbol{A}}, ||\boldsymbol{p_a} - \boldsymbol{p_b}||_2 < r\}$, $\phi(\cdot)$ denotes the functional expansions, and $H(\cdot)$ is a non-linear function based on 3D GNN models.

However, we argue that predicting or approximating the energy with Eqn. (4) is a suboptimal solution. Actually, compared with Eqn. (3), which is physics-principled, there exist non-negligible approximation errors in Eqn. (4). First, Eqn. (4) captures the interatomic interactions based on interatomic distances, while the energy can be more accurately approximated by a function of interatomic potentials as in Eqn. (3). Though according to Sec. 2.2, interatomic potentials themselves are also functions of distances, we argue that directly using functional expansions of distances is not the best solution to crystal energy prediction. The commonly used functional expansions in existing methods, such as RBF $\phi(\cdot)$, have different mathematical forms from potentials defined in Sec. 2.2. Intuitively, this poses more challenges to 3D GNN models since they need to learn a mapping from $\phi(\cdot)$ to the energy $\boldsymbol{E}$, while the energy $\boldsymbol{E}$ is not a direct function of $\phi(\cdot)$. Hence, we argue that directly employing the physics-principled potential functions instead of $\phi(\cdot)$ as input features is more suitable for crystal energy prediction.

Second, different from Eqn. (3), Eqn. (4) does not capture the complete set of interatomic interactions because the summation set $\mathcal{N}_r(\boldsymbol{a})$ of atoms $\boldsymbol{b}$ is constrained to be the atoms whose distances to the atom $\boldsymbol{a}$ are smaller than $r$. This can lead to a significant approximation error due to ignoring long-range interatomic potentials, i.e., interatomic potentials between distant atoms. Different from molecules with finite structures, long-range interatomic potentials cannot be ignored for crystals with infinite structures. By the first principles in physics, interatomic potentials decay algebraically when pairwise interatomic distances become larger. Hence, for a finite structure like molecules, the potentials from atoms that are far away from a given atom is limited and can be ignored. However, long-range interatomic potentials have a significant influence on a given atom in the infinite crystal structure. Taking Coulomb potentials as an example, assuming that we are given a 1D crystal structure where there is only one atom repeating itself with Euclidean distance of 1, each atom has only one unit of charge, and the total energy is simply the sum of all interatomic Coulomb potentials. As defined in Sec. 2.2, Coulomb potential energy $V(\boldsymbol{a}, \boldsymbol{b})$ between atoms $\boldsymbol{a}$ and $\boldsymbol{b}$ satisfies $V(\boldsymbol{a}, \boldsymbol{b}) \propto 1/d$, where $d$ is the distance between atoms $\boldsymbol{a}$ and $\boldsymbol{b}$. Considering the

Coulomb potentials between a given atom and all other atoms, the total potential $\widetilde{V}$ of them satisfies $\widetilde{V} \propto \sum_{n=1}^{\infty} 1/n$. If only the pairwise atoms within the distance threshold $r$ are considered, an infinite error of energy calculation is introduced. Specifically, the smallest possible prediction error $\Delta V$ satisfies $\Delta V \propto \sum_{n=\lfloor r+1 \rfloor}^{\infty} 1/n$, which is infinite. In other words, ignoring interatomic Coulomb potentials between atoms with distances larger than $r$ will cause a significant prediction error of the total energy. We can observe from this example that the failure to capture complete interatomic potentials due to the use of radius graphs is a key factor that prevents accurate energy prediction in existing GNN-based methods.

## 3.3 Message Passing with Complete Interatomic Potentials

It follows from the analysis in Sec. 3.2 that major limitations of existing deep learning methods for crystal representation learning lie in (1) not making predictions from physics-principled interatomic potentials, and (2) not considering complete interatomic interactions. To overcome these limitations, we propose to explicitly use complete interatomic potential summations in GNN models. Since our proposed method is tightly related to potentials, we name it PotNet.

By reformulating Eqn. (3), our PotNet incorporates the crystal energy computation with complete interatomic potentials into the message passing scheme of GNN models. For any material structure $M = (A, L)$, we can rewrite the definition of its complete structure $\widetilde{A}$ in Eqn. (1) as

$$\widetilde{A} = \bigcup_{k \in \mathbb{Z}^3} A^k = \bigcup_{k \in \mathbb{Z}^3} \bigcup_{b \in A} \{b^k\} = \bigcup_{b \in A} \bigcup_{k \in \mathbb{Z}^3} \{b^k\} = \bigcup_{b \in A} A_b, \tag{5}$$

where $A_b = \bigcup_{k \in \mathbb{Z}^3} \{b^k\}$ denotes the set of atoms containing the atom $b$ from the unit cell $A$ and all its periodically repeated duplicates in the complete crystal structure. With Eqn. (5), we can reformulate Eqn. (3) as

$$\begin{aligned}
E(M) &= \sum_{a \in A} \left[ \frac{1}{2} \sum_{b \in A} \sum_{c \neq a, c \in A_b} V(a, c) + G \left( \sum_{b \in A} \sum_{c \neq a, c \in A_b} V(a, c) \right) \right] \\
&= \sum_{a \in A} \left[ \frac{1}{2} \sum_{b \in A} S(a, b) + G \left( \sum_{b \in A} S(a, b) \right) \right],
\end{aligned} \tag{6}$$

where the infinite potential summation $S(a, b) = \sum_{c \neq a, c \in A_b} V(a, c)$ denotes the sum of the interatomic potentials from the atom $b$ together with its all periodic duplicates to the atom $a$. Eqn. (6) can be integrated into the message passing scheme of GNN models. Specifically, we can create a graph $G$ for $M = (A, L)$, where each atom in the unit cell $A$ corresponds to a node in the graph. For any two nodes $u, v$ in the graph, there is an edge connecting them, and every node $u$ in the graph is also connected to itself by a self-loop edge. If we consider the infinite potential summation $S(a, b)$ as the feature of the edge from node $b$ to $a$, we can use the message passing based non-linear neural network model in GNN to fit the function $\frac{1}{2} \sum_{b \in A} S(a, b) + G \left( \sum_{b \in A} S(a, b) \right)$.

Based on this design of directly using interatomic potentials as edge features, our PotNet employs a GNN model with multiple message passing layers on the graph $G$ to predict the crystal energy of $M$. The computational process of the $\ell$-th message passing layer for the node $a$ can be described as

$$h_a^{(\ell)} = g_\varphi \left( h_a^{(\ell-1)}, \sum_{b \in A} f_\theta \left( h_a^{(\ell-1)}, h_b^{(\ell-1)}, \sum_{b \in A} S(a, b) \right) \right), \tag{7}$$

where $h_a^{(\ell)}$ denotes the embedding vector of node $a$ generated from the $\ell$-th message passing layer, $h_a^{(0)}$ is initialized to the atom feature vector of the atom $a$, and $g_\varphi(\cdot), f_\theta(\cdot)$ are both neural network models with trainable parameters $\varphi$ and $\theta$, respectively. Here, the model $f_\theta$ plays the role of capturing information from both atomic features and complete interatomic potentials. Detailed information about model architectures of $f_\theta$ and $g_\varphi$ is provided in Appendix D.1. Note that our PotNet is actually a 3D GNN model even though 3D geometric information is not explicitly involved in Eqn. (7). This is because the edge feature $S(a, b)$ is related to potential functions, and by Sec. 2.2 we know that they are computed from interatomic distances. In other words, PotNet can be considered to encode 3D geometric information with potential functions, though our direct motivation of using potential functions comes from the physical modeling of crystal energy.

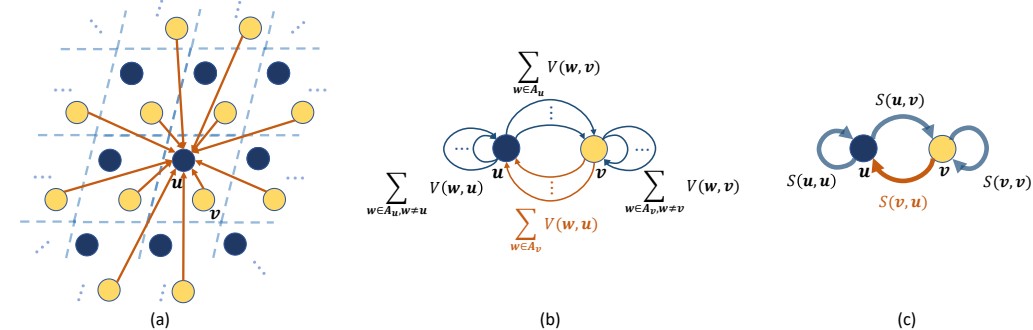

Figure 1: Schematic illustrations of how complete interatomic itneractions are captured in PotNet. Note that PotNet models 3D crystals while we have 2D illustration for simplicity. (a) An example crystal in which each unit cell contains two atoms $u$ and $v$. In PotNet, the potentials between all pairs of atoms are captured. For simplicity, we only show the potentials from all $v$ atoms to a $u$ atom. (b) The complete set of potentials in (a) can be grouped into four categories, including $u \to v$, $v \to u$, $u \to u$, and $v \to v$. (c) We propose to compute an approximate summation for each category of potentials.

Intuitively, the message passing process in Eqn. (7) over the graph $G$ can be considered as a general case of employing a radius graph where the distance threshold $r$ goes to infinity, i.e., $r \to +\infty$. In this case, as shown in Fig. 1(a), for any atom in the crystal, all the other atoms in the complete crystal structure have been included to interact with it. If we follow the radius graph construction process in the previous methods (Xie & Grossman, 2018; Chen et al., 2019; Louis et al., 2020; Choudhary & DeCost, 2021), we obtain a graph $\widetilde{G}$ in which there exist an infinite number of edges between every pair of nodes. However, PotNet simplifies this complicated graph $\widetilde{G}$ to the graph $G$ in which only one edge exists between every node pair. Specifically, PotNet directly models interatomic interactions as potentials and for any two nodes in $\widetilde{G}$, PotNet aggregates all edges between them to a single edge by the use of infinite potential summation $S(\boldsymbol{a}, \boldsymbol{b})$ (see Fig. 1(b)). In other words, PotNet provides an effective solution that enables GNN models to capture complete interatomic interactions through the use of infinite potential summations.

## 3.4 EFFICIENT COMPUTATION OF INFINITE POTENTIAL SUMMATION

Although we have effectively incorporated infinite potential summations into the message passing based GNN models, the computation of these infinite potential summations is not easy. Basically, there are two challenges to achieve accurate and efficient computation of the infinite potential summations. For accuracy, the computation algorithm needs to have provable error bounds. For efficiency, it needs to be fast for scalable GNN training and fast crystal property prediction. To tackle these two challenges, we derive a fast approximation algorithm for infinite potential summations based on the Ewald summation method (Ewald, 1921). Specifically, we unify the summations of three infinite potentials between atom $\boldsymbol{a}$ and all duplicated positions of atom $\boldsymbol{b}$ into an integral form such that the Ewald summation method can be applied for efficient implementation in PotNet (Fig. 1(c)). The key idea of the Ewald summation is that a slowly converging summation in the real space is guaranteed to be converted into a quickly converging summation in the Fourier space (Woodward, 2014). Based on that, the Ewald summation method divides a summation into two parts. One part has a quicker converging rate in the real space than the original summation. The other "slower-to-converge" part is transformed into the Fourier space and becomes quickly convergent. In our method, the Ewald summation method is used with the infinite summations by dividing the integral into two parts, including one part that converges fast in the Fourier space and another part that converges fast in the real space, to obtain fast approximations with provable error bounds.

To apply the fast approximation algorithm of infinite summations proposed by Ewald (1921), a unified integral view of infinite potential summations is needed. Following notations in Sec. 2 and 3.3, we denote the positions of the atoms in the set $\boldsymbol{A_b}$ as $\boldsymbol{P_b} = \{\boldsymbol{p_b^k} \,|\, \boldsymbol{p_b^k} = \boldsymbol{p_b} + \boldsymbol{Lk}, \boldsymbol{k} \in \mathbb{Z}^3\}$. The Euclidean distances between the atom $\boldsymbol{a}$ and all atoms in $\boldsymbol{A_b}$ can be represented as $\{d \,|\, d = \|\boldsymbol{p_b} + \boldsymbol{Lk} - \boldsymbol{p_a}\|, \boldsymbol{k} \in \mathbb{Z}^3\}$. As described in Sec. 3.3, we simplify the Coulomb potential function as $V_{\text{Coulomb}}(\boldsymbol{a}, \boldsymbol{b}) = -\epsilon_1/d(\boldsymbol{a}, \boldsymbol{b})$, where $\epsilon_1$ is a hyperparameter scaling the potential. Based

Table 1: Comparison between PotNet and other baselines in terms of test MAE on the Materials Project dataset. To make the comparison clear and fair, we retrain baseline methods using the same dataset settings. We also show results from original papers in parentheses and mark missing results as -. The best results are shown in **bold** and the second best results are shown with underlines.

| Method | Formation Energy eV/atom | Band Gap eV | Bulk Moduli log(GPa) | Shear Moduli log(GPa) |
|---|---|---|---|---|
| CGCNN | 0.031 (0.039) | 0.292 (0.388) | 0.047 (0.054) | 0.077 (0.087) |
| SchNet | 0.033 (0.035) | 0.345 (-) | 0.066 (-) | 0.099 (-) |
| MEGNET | 0.030 (0.028) | 0.307 (0.33) | 0.051 (0.050) | 0.088 (0.079) |
| GATGNN | 0.033 (0.039) | 0.280 (0.31) | 0.045 | 0.075 |
| ALIGNN | 0.0221 | 0.0218 | 0.051 (-) | 0.078 (-) |
| PotNet | **0.0196** | **0.0204** | **0.042** | **0.069** |

on that, we can represent Coulomb potentials from all atoms in $\boldsymbol{A_b}$ to the atom $\boldsymbol{a}$ as $\{-\frac{\epsilon}{d} \mid d = \|\boldsymbol{p_b} + \boldsymbol{Lk} - \boldsymbol{p_a}\|, \boldsymbol{k} \in \mathbb{Z}^3\}$. Similarly, London dispersion potentials from all atoms in $\boldsymbol{A_b}$ to the atom $\boldsymbol{a}$ can be represented as $\{-\frac{\epsilon'}{d^6} \mid d = \|\boldsymbol{p_b} + \boldsymbol{Lk} - \boldsymbol{p_a}\|, \boldsymbol{k} \in \mathbb{Z}^3\}$. It is worth noting that Coulomb potentials and London dispersion potentials can be represented in a unified view as $\{\frac{\text{constant}}{d^p} \mid d = \|\boldsymbol{v_{ab}} + \boldsymbol{Lk}\|, \boldsymbol{v_{ab}} = \boldsymbol{p_b} - \boldsymbol{p_a}, \boldsymbol{k} \in \mathbb{Z}^3\}$, where $p$ is a positive number. And we represent Pauli potentials from all positions of atom $j$ to atom $i$ as $\{e^{-\alpha d} \mid d = \|\boldsymbol{p_b} + \boldsymbol{Lk} - \boldsymbol{p_a}\|, \boldsymbol{k} \in \mathbb{Z}^3\}$. We provide detailed proofs in Appendix C.1 that the summations of these three potentials can be unified in an integral form as

$$S(\boldsymbol{a}, \boldsymbol{b}) = D \sum_{\boldsymbol{k} \in \mathbb{Z}^3} \int_0^\infty t^{C-1} e^{-A\pi|\boldsymbol{Lk} + \boldsymbol{v_{ab}}|^2 t - \frac{B}{t}} dt, \qquad (8)$$

where $A, B, C, D$ are constants derived from the corresponding specific potential forms. We then apply the Ewald summation method (Ewald, 1921) to Eqn. (8) and split it into two parts as

$$\begin{aligned} S(\boldsymbol{a}, \boldsymbol{b}) =& D \sum_{\boldsymbol{k} \in \mathbb{Z}^3} \int_0^1 t^{C-1} e^{-A\pi|\boldsymbol{Lk} + \boldsymbol{v_{ab}}|^2 t - \frac{B}{t}} dt + D \sum_{\boldsymbol{k} \in \mathbb{Z}^3} \int_1^\infty t^{C-1} e^{-A\pi|\boldsymbol{Lk} + \boldsymbol{v_{ab}}|^2 t - \frac{B}{t}} dt \\ =& S_{\text{Fourier}}(\boldsymbol{a}, \boldsymbol{b}) + S_{\text{direct}}(\boldsymbol{a}, \boldsymbol{b}), \end{aligned} \qquad (9)$$

where $S_{\text{direct}}$ denotes the part that converges fast in real space, and $S_{\text{Fourier}}$ denotes the other part that quickly converges in Fourier space when the total summation converges as shown by Ewald (1921). Based on this, we further show that $S_{\text{direct}}$ and $S_{\text{Fourier}}$ can be expressed as summations of incomplete Bessel functions $K_\nu(x, y)$ in Appendix C.2, and the approximation error can be analyzed and shown to be bounded. Note that for London dispersion potentials and Pauli potentials, the transformed summations of incomplete Bessel functions can be directly approximated. However, for Coulomb potentials, the direct summation diverges. Concretely, when $p > 3$ for potentials of form $\{\frac{\text{constant}}{d^p} \mid d = \|\boldsymbol{v_{ab}} + \boldsymbol{Lk}\|, \boldsymbol{v_{ab}} = \boldsymbol{p_b} - \boldsymbol{p_a}, \boldsymbol{k} \in \mathbb{Z}^3\}$, the corresponding potential summation converges, and when $p \leq 3$, the corresponding potential summation diverges. To tackle this problem, we follow previous mathematical derivations (Harris, 2008; Slevinsky & Safouhi, 2010; Jones, 2007) and use the analytically continued incomplete Bessel functions to approximate infinite summations of Coulomb potentials as shown in Appendix C.3. We then provide detailed mathematical proofs that the summations of incomplete Bessel functions are convergent and can be approximated with an error bounded by the Gaussian Lattice Sum in Appendix B.2. Detailed implementation of the proposed summation algorithm can be found in Appendix C.4. It is worth noting that PotNet is the first method to use the incomplete Bessel function to compute the potential summations and is able to compute the Pauli potential summation, while previous methods (Crandall, 1998; Lee & Cai, 2009; Nestler et al., 2015) cannot achieve this. Also, we are able to compute other interatomic potential summations including Lennard-Jones potential, Morse potential, and screened Coulomb potential by using our method as shown in Appendix C.5.

Table 2: Comparison between PotNet and other baselines in terms of test MAE on JARVIS dataset. The best results are shown in **bold** and the second best results are shown with underlines.

| Method | Formation Energy eV/atom | Bandgap(OPT) eV | Total energy eV/atom | Bandgap(MBJ) eV | Ehull eV |
|---|---|---|---|---|---|
| CFID | 0.14 | 0.30 | 0.24 | 0.53 | 0.22 |
| CGCNN | 0.063 | 0.20 | 0.078 | 0.41 | 0.17 |
| SchNet | 0.045 | 0.19 | 0.047 | 0.43 | 0.14 |
| MEGNET | 0.047 | 0.145 | 0.058 | 0.34 | 0.084 |
| GATGNN | 0.047 | 0.17 | 0.056 | 0.51 | 0.12 |
| ALIGNN | 0.0331 | 0.142 | 0.037 | 0.31 | 0.076 |
| PotNet | **0.0308** | **0.136** | **0.034** | **0.28** | **0.050** |

## 4 EXPERIMENTAL STUDIES

### 4.1 EXPERIMENTAL SETUP

We conduct experiments on two material benchmark datasets, including The Materials Project and JARVIS. Baseline methods include CFID (Choudhary et al., 2018), SchNet (Schütt et al., 2017), CGCNN (Xie & Grossman, 2018), MEGNET (Chen et al., 2019), GATGNN (Louis et al., 2020), and ALIGNN (Choudhary & DeCost, 2021). Unless otherwise specified, we report the results reported by referred papers or provided by original authors. All PotNet models are trained using the Adam (Kingma & Ba, 2014) optimizer with weight decay (Loshchilov & Hutter, 2017) and one cycle learning rate scheduler (Smith & Topin, 2019) with a learning rate of 0.001, training epoch of 500, and batch size of 64. We use Pytorch to implement our models. For all tasks on two benchmark datasets, we use one NVIDIA RTX A6000 48G GPU for computing. Other detailed configurations of PotNet for different tasks are provided in Appendix D.1.

To capture the global infinite-range interactions without losing details of local interactions, PotNet uses both local and infinite crystal graphs. Specifically, for the local crystal graph, we use the radius crystal graph proposed by CGCNN but replace Euclidean distances with interatomic potentials for edge features. To be concrete, because the influences of London dispersion potentials and Pauli potentials are limited and can be ignored when only considering nearby regions, we only use the Coulomb potentials in the radius crystal graph. The infinite crystal graph is constructed as described in Sec. 3.3, where Coulomb potentials, London dispersion potentials and Pauli potentials are used.

### 4.2 EXPERIMENTAL RESULTS

**The Materials Project**. We first evaluate PotNet on The Materials Project-2018.6.1, which is a widely used large-scale material benchmark with 69239 crystals. We follow previous works (Xie & Grossman, 2018; Chen et al., 2019; Choudhary & DeCost, 2021; Louis et al., 2020) and use four crystal properties including formation energy, band gap, bulk moduli, and shear moduli. We notice that previous works (Xie & Grossman, 2018; Chen et al., 2019; Choudhary & DeCost, 2021; Louis et al., 2020; Schütt et al., 2017) compare with each other using different versions of splitting training, evaluation, and testing datasets with different random seeds. For instance, *the original CGCNN paper only uses 28046 training samples for formation energy prediction, resulting in the original result of 0.039 as shown in Table 1*. To make the comparisons fair, we follow the settings of the previous state-of-the-art (SOTA) ALIGNN (Choudhary & DeCost, 2021) for tasks including formation energy and band gap prediction, and retrain all other baselines using the same dataset setting for these two tasks. For bulk and shear moduli, we follow the dataset setting of GATGNN (Louis et al., 2020) which has the best prediction performances for these two tasks and retrained all other baselines. Detailed configurations are also shown in Appendix D.2. We present our results in Table 1, where PotNet consistently outperforms other SOTA methods on all four tasks by large margins.

**JARVIS Dataset**. We then evaluate PotNet on JARVIS, a newly released benchmark dataset proposed by Choudhary et al. (2020) with 55722 crystals. We evaluate PotNet on five crystal property prediction tasks including formation energy, bandgap (OPT), bandgap (MBJ), total energy, and Ehull. We follow ALIGNN and use the same training, validation, and test sets for these tasks. Since there are missing results for baseline methods, we retrain corresponding baseline methods follow-

ing the same settings with ALIGNN as discussed in Appendix D.2. As shown in Table 2, PotNet achieves the best performances on all five tasks consistently by significant margins.

**Efficiency of PotNet**. Beyond the superior modeling capacity for crystals, our PotNet is faster and more efficient than ALIGNN. To demonstrate the efficiency of PotNet, we compare PotNet with ALIGNN in terms of training time per epoch, total training time, and inference time for the task of JARVIS formation energy prediction. From Table 3, PotNet is four times faster in terms of total training time and inference time compared with ALIGNN.

Table 3: Model complexity and runtime compared with ALIGNN on JARVIS formation energy.

| Method | #Params | Time/Epoch | Total Training Time | Total Testing Time |
|--------|---------|------------|---------------------|--------------------|
| ALIGNN | 15.4 MB | 327 s | 27.3 h | 156 s |
| PotNet | **7.9MB** | **48 s** | **6.7 h** | **41 s** |

We also analyze the time cost of infinite summations as shown in Table 4. We calculate the infinite summations during preprocessing. Unlike previous methods including ALIGNN, we need to calculate the complete potential set besides constructing graphs. Therefore, we do require more computing time for preprocessing. However, for a single material, the preprocessing time of our method is at the level of milliseconds. Even if we additionally compute the infinite potential summations, the preprocessing time of our method and ALIGNN is still within the same order of magnitude. If considering both preprocessing time and model inference time for single material screening, we have a faster inference speed than ALIGNN as illustrated in the table. Overall, the computational cost of our method is reasonable.

Table 4: Preprocessing time compared with ALIGNN on the JARVIS dataset. The first and second columns show the preprocessing time on the whole JARVIS dataset with 55722 crystals. The third column denotes the mean inference time considering both preprocessing and model time.

| Method | Preprocessing Time/Crystal | Total Preprocessing Time | Inference/Crystal |
|--------|----------------------------|--------------------------|-------------------|
| ALIGNN | **2.7 ms** | **152 s** | 30 ms |
| PotNet | 10.8 ms | 603 s | **18 ms** |

### 4.3 Ablation Studies

In this section, we demonstrate the importance of two core components of PotNet, including interaction modeling using potentials and infinite summation of potentials for crystal prediction. We conduct experiments on the JARVIS formation energy task, and use test MAE as the evaluation metric.

**Interaction Modeling using Potentials**. We demonstrate the importance of interaction modeling using potentials by replacing potentials with Euclidean distances used by previous works in our PotNet with exactly the same model architecture. Specifically, we denote PotNet with only local crystal graph as the base model. We use 'Base + Euclidean' to represent the base model with Euclidean distances as edge features and 'Base + Potential' to represent the base model using Coulomb potentials as edge features. As shown in Table 5, by replacing Euclidean distances with Coulomb potentials, PotNet without considering infinite potential summation already obtains a significant performance gain from 0.0363 to 0.0318, revealing the importance of interaction modeling using potentials in PotNet.

Table 5: Ablation studies for the effects of adding Coulomb potentials and infinite summation.

| Method | JARVIS Formation Energy |
|--------|-------------------------|
| Base + Euclidean | 0.0363 |
| Base + Potential | 0.0318 |
| Base + Potential + Infinite | 0.0308 |

**Infinite Summation of Potentials**. The importance of infinite summation of potentials is demonstrated by comparing the previous base models with 'Base + Potential + Infinite', denoting the full PotNet model with infinite summation in infinite crystal graph. It can be seen from Table 5 that by using infinite crystal graphs introduced in Sec. 3.3, the global repeating patterns of crystal structures are captured, resulting in a performance gain from 0.0318 to 0.0308 for formation energy prediction.

## 5 Conclusion

We study the problem of how to capture infinite-range interatomic potentials in crystal property prediction directly. As a radical departure from prior methods that only consider nearby atoms, we develop a new GNN, PotNet, with the message passing scheme that takes efficient approximations to capture the complete set of potentials among all atoms. Experiments show that the use of complete potentials leads to consistent performance improvements. Altogether, our work provides a theoretically principled and practically effective framework for crystal modeling.

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

## A  GAUSSIAN LATTICE SUM

The Gaussian Lattice Sum (Bétermin et al., 2021) computes the summation of Gaussian functions centered at the points given by a shifted lattice, formally defined as

$$G_\Omega(\boldsymbol{L}, \boldsymbol{v}, c) = \sum_{\boldsymbol{k} \in \mathbb{Z}^d} e^{-c|\boldsymbol{L}\boldsymbol{k}+\boldsymbol{v}|^2}, \tag{10}$$

where $\boldsymbol{L} \in \mathbb{R}^{d \times d}$ is the lattice matrix, $\boldsymbol{v} \in \mathbb{R}^d$ is a vector inside a unit cell and $c \in \mathbb{R}^+$ is a prefixed constant. One of the characteristics of the Gaussian Lattice Sum is that the term $e^{-c|\boldsymbol{L}\boldsymbol{k}+\boldsymbol{v}|^2}$ rapidly decays as $\boldsymbol{k}$ becomes large, resulting fast convergence of $G_\Omega(\boldsymbol{L}, \boldsymbol{v}, c)$. As shown by Deconinck et al. (2004), for $R \in \mathbb{R}^+$, we have

$$\sum_{\boldsymbol{k} \in \mathbb{Z}^d, \sqrt{c}|\boldsymbol{L}\boldsymbol{k}+\boldsymbol{v}| \geq R} e^{-c|\boldsymbol{L}\boldsymbol{k}+\boldsymbol{v}|^2} \leq \frac{d}{2}\left(\frac{2}{\rho}\right)^d \Gamma\left(\frac{d}{2}, \left(R - \frac{\rho}{2}\right)^2\right), \tag{11}$$

where $\Gamma(z, x) = \int_x^\infty t^{z-1}e^{-t}dt$ is the incomplete Gamma function and $\rho = \min\{\sqrt{c}|\boldsymbol{L}\boldsymbol{k}| \,|\boldsymbol{k} \in \mathbb{Z}^d, \boldsymbol{k} \neq \boldsymbol{0}\}$. We can obtain the upper bound of the Gaussian Lattice Sum by setting $R = 0$ such that

$$\sum_{\boldsymbol{k} \in \mathbb{Z}^d} e^{-c|\boldsymbol{L}\boldsymbol{k}+\boldsymbol{v}|^2} \leq \frac{d}{2}\left(\frac{2}{\rho}\right)^d \Gamma\left(\frac{d}{2}, \left(\frac{\rho}{2}\right)^2\right). \tag{12}$$

## B  INCOMPLETE BESSEL FUNCTION

The incomplete Bessel function (Harris, 2008) is defined as

$$K_\nu(x, y) = \int_1^\infty t^{-\nu-1}e^{-xt-y/t}dt, \tag{13}$$

where $x, y \in \mathbb{R}$, $x \geq 0$, $y \geq 0$, and $\nu > 0$. In existing studies (Harris, 2008; Slevinsky & Safouhi, 2010; Jones, 2007), $K_\nu(x, y)$ can be analytically continued to $\nu \in \mathbb{R}$. In this work, we follow these studies and also use the analytic continuation for our calculations, i.e., we also consider $\nu \in \mathbb{R}$.

### B.1  FAST APPROXIMATION OF THE INCOMPLETE BESSEL FUNCTION

Computing the incomplete Bessel function is extremely challenging as there does not exist any explicit closed-form solution. In this work, we investigate a fast approximation of the incomplete Bessel function. Specifically, we adopt the algorithm in the work by Slevinsky & Safouhi (2010), where the incomplete Bessel function $K_\nu(x, y)$ is approximated by the $G_n^{(m)}$ transformation with the linear time complexity of $O(n)$, and $n$ is the number of iterations. As shown by Levin & Sidi (1981), the approximation $G_n^{(m)}$ to $\int_0^\infty f(t)dt$ is given as the solution of

$$\frac{d^l}{dx^l}\left\{G_n^{(m)} - \int_0^x f(t)dt - \sum_{k=0}^{m-1} x^{\sigma_k}f^{(k)}(x)\sum_{i=0}^{n-1}\frac{\beta_{i,k}}{x^i}\right\} = 0, \tag{14}$$

where $\beta_{i,k}$ and $G_n^{(m)}$ are unknowns, $\frac{d^l}{dx^l}G_n^{(m)} = 0, \forall l > 0$, $\sigma_k = min(s_k, k+1)$, and $s_k$ is the largest of $s \in \mathbb{Z}$ such that $\lim_{x \to \infty} x^s f^{(k)}(x) = 0$ for $k = 0, 1, \cdots, m-1$. Slevinsky & Safouhi (2010) proved that the incomplete Bessel function satisfies Eqn. (14) with $m = 1$, through which we can obtain

$$G_n^{(1)} - \int_0^x f(t)dt = x^{\sigma_0}f(x)\sum_{i=0}^{n-1}\frac{\beta_{i,k}}{x^i}. \tag{15}$$

To eliminate all the unknowns $\beta_{i,k}$, Slevinsky & Safouhi (2010) applied the $(x^2 \frac{d}{dx})$ operator $n$ times such that

$$(x^2 \frac{d}{dx})^n \left[ \frac{G_n^{(1)} - \int_0^x f(t)dt}{x^{\sigma_0} f(x)} \right] = 0. \tag{16}$$

By doing this, we obtain

$$G_n^{(1)} = \frac{(x^2 \frac{d}{dx})^n (\frac{\int_0^x f(t)dt}{x^{\sigma_0} f(x)})}{(x^2 \frac{d}{dx})^n (\frac{1}{x^{\sigma_0} f(x)})} = \frac{\mathcal{N}_n(x)}{\mathcal{D}_n(x)}, \tag{17}$$

in which we can have

$$\mathcal{N}_n(x) = (x^2 \frac{d}{dx}) \mathcal{N}_{n-1}(x) \quad \text{and} \quad \mathcal{D}_n(x) = (x^2 \frac{d}{dx}) \mathcal{G}_{n-1}(x) \tag{18}$$

with

$$\mathcal{N}_0(x) = \frac{\int_0^x f(t)dt}{x^{\sigma_0} f(x)} \quad \text{and} \quad \mathcal{D}_0(x) = \frac{1}{x^{\sigma_0} f(x)}. \tag{19}$$

This leads to a recursive algorithm to approximate $G_n^{(1)}$. To compute the incomplete Bessel function $K_\nu(x, y)$, Slevinsky & Safouhi (2010) investigated the following property

$$K_\nu(x, y) + x^\nu \int_0^x t^{-\nu-1} e^{-t-\frac{xy}{t}} dt = x^\nu \int_0^\infty t^{-\nu-1} e^{-t-\frac{xy}{t}} dt, \tag{20}$$

in which the term $\int_0^\infty t^{-\nu-1} e^{-t-\frac{xy}{t}} dt$ can be approximated by $G_n^{(1)}$. Therefore, to approximate $K_\nu(x, y)$ we have

$$\begin{aligned}
\tilde{G}_n^{(1)} &= x^\nu (G_n^{(1)} - \int_0^x t^{-\nu-1} e^{-t-\frac{xy}{t}} dt) \\
&= x^\nu \frac{\mathcal{N}_n(x) - \mathcal{D}_n(x) \int_0^x t^{-\nu-1} e^{-t-\frac{xy}{t}} dt}{\mathcal{D}_n(x)} \\
&= x^\nu \frac{\sum_{r=1}^n \binom{n}{r} \mathcal{D}_{n-r}(x) (x^2 \frac{d}{dx})^{r-1} (x^{-\nu+1} e^{-x-y})}{\mathcal{D}_n(x)} \\
&= \frac{\tilde{\mathcal{N}}_n(x)}{\mathcal{D}_n(x)}.
\end{aligned} \tag{21}$$

As a result, we obtain the approximation $\tilde{G}_n^{(1)}$ to $K_\nu(x, y)$ by recursively solving $\tilde{\mathcal{N}}_n(x)$ and $\mathcal{D}_n(x)$ (Gaudreau et al., 2012). The detailed expressions of $\tilde{\mathcal{N}}_n(x)$ and $\mathcal{D}_n(x)$ are given in Slevinsky & Safouhi (2010). In addition, we follow Nestler et al. (2015) to further optimize the approximation of the incomplete Bessel function when $\nu = 0$ and $x, y$ are both small, e.g. $x^2 + y^2 < 1$. In this case, the remaining part of the Taylor expansion of $K_0(x, y)$ is small and we can approximate $K_0(x, y)$ by the first $m$ terms of Taylor series such that

$$K_0(x, y) \approx \sum_{n=0}^m \frac{(-1)^n}{n!} x^n y^n \Gamma(-n, x). \tag{22}$$

The detailed error bound by this expansion is shown in Nestler et al. (2015).

## B.2 Convergence of Incomplete Bessel Function Summation

We define the summation of incomplete Bessel functions on a lattice as

$$\sum_{\boldsymbol{k} \in \mathbb{Z}^d} K_\nu(\alpha |\boldsymbol{Lk} + \boldsymbol{v}|^2 + \gamma, \beta), \tag{23}$$

where $\nu, \alpha, \beta, \gamma \in \mathbb{R}$ are constants, $\alpha > 0$, $\beta \geq 0$, $\gamma \geq 0$, $\boldsymbol{v} \in \mathbb{R}^d$ is a vector inside a unit cell and $\boldsymbol{L} \in \mathbb{R}^{d \times d}$ is the lattice matrix. We aim to prove the summation of incomplete Bessel functions is convergent and can be approximated with an error bounded by the Gaussian Lattice Sum introduced in Appendix A.

*Proof.* The incomplete Bessel function has an upper bound such that

$$\left|K_\nu(x,y)\right| = \left|\int_1^\infty t^{-\nu-1}e^{-xt-y/t}dt\right| \le \left|\int_1^\infty t^{-\nu-1}e^{-xt}dt\right| = \left|x^{-\nu}\Gamma(-\nu,x)\right|, \quad (24)$$

where $x > 0$ and $\Gamma$ is the incomplete Gamma function described in Appendix A. Based on this, we obtain

$$\left|K_\nu(\alpha|\boldsymbol{Lk}+\boldsymbol{v}|^2+\gamma,\beta)\right| \le \left|\frac{\Gamma(-\nu,\alpha|\boldsymbol{Lk}+\boldsymbol{v}|^2+\gamma)}{(\alpha|\boldsymbol{Lk}+\boldsymbol{v}|^2+\gamma)^\nu}\right|, \quad (25)$$

where $\alpha|\boldsymbol{Lk}+\boldsymbol{v}|^2+\gamma > 0$. As shown by Borwein & Chan (2009), $\left|\frac{\Gamma(z,x)}{x^z}\right|$ has an upper bound such that

$$\left|\frac{\Gamma(z,x)}{x^z}\right| = \left|e^{-x}\int_0^\infty e^{-xs}(1+s)^{z-1}ds\right|$$
$$\le \begin{cases} \frac{e^{-x}}{x-z+1}, & z > 1 \\ \frac{e^{-x}}{x}, & z \le 1 \end{cases}, \quad (26)$$

where $z \in \mathbb{R}$, $x \in \mathbb{R}$ and $x > 0$. For a prefixed value $R \in \mathbb{R}$, $R^2 > -\nu$, $R^2 > 1$ and $R^2 > \gamma$, we have

$$\epsilon(R) = \sum_{\boldsymbol{k}\in\mathbb{Z}^d, \alpha|\boldsymbol{Lk}+\boldsymbol{v}|^2+\gamma\ge R^2} K_\nu(\alpha|\boldsymbol{Lk}+\boldsymbol{v}|^2+\gamma,\beta)$$
$$\le \sum_{\boldsymbol{k}\in\mathbb{Z}^d, \alpha|\boldsymbol{Lk}+\boldsymbol{v}|^2+\gamma\ge R^2} |K_\nu(\alpha|\boldsymbol{Lk}+\boldsymbol{v}|^2+\gamma,\beta)| \quad (27)$$
$$\le \sum_{\boldsymbol{k}\in\mathbb{Z}^d, \alpha|\boldsymbol{Lk}+\boldsymbol{v}|^2+\gamma\ge R^2} e^{-\alpha|\boldsymbol{Lk}+\boldsymbol{v}|^2-\gamma}$$
$$\le e^{-\gamma}G_\Omega(\boldsymbol{L},\boldsymbol{v},\alpha),$$

where $G_\Omega(\boldsymbol{L},\boldsymbol{v},\alpha)$ is the Gaussian Lattice Sum as described in Appendix A. Therefore, the incomplete Bessel function summation can be divided into two parts such that

$$\sum_{\boldsymbol{k}\in\mathbb{Z}^d} K_\nu(\alpha|\boldsymbol{Lk}+\boldsymbol{v}|^2+\gamma,\beta) = \sum_{\boldsymbol{k}\in\mathbb{Z}^d, \alpha|\boldsymbol{Lk}+\boldsymbol{v}|^2+\gamma\le R^2} K_\nu(\alpha|\boldsymbol{Lk}+\boldsymbol{v}|^2+\gamma,\beta) + \epsilon(R), \quad (28)$$

where the first part is a finite part inside an ellipsoid with a size of $\sqrt{R^2/\alpha - \gamma/\alpha}$, and the second part $\epsilon(R)$ is bounded by the Gaussian Lattice Sum $G_\Omega(\boldsymbol{L},\boldsymbol{v},\alpha)$ which is convergent. Thus, the incomplete Bessel function summation is convergent. Consequently, to approximate the incomplete Bessel function summation, we choose to evaluate the summation inside an ellipsoid with the size of $\sqrt{R^2/\alpha - \gamma/\alpha}$ for a prefixed $R \in \mathbb{R}$, such that $R^2 > -\nu$, $R^2 > 1$ and $R^2 > \gamma$. Then the error $\epsilon(R)$ is bounded by Gaussian Lattice Sum $G_\Omega(\boldsymbol{L},\boldsymbol{v},\alpha)$. We can further bound the error by the inequality (11) of Gaussian Lattice Sum as introduced in Appendix A such that

$$\epsilon(R) \le \sum_{\boldsymbol{k}\in\mathbb{Z}^d, \sqrt{\alpha}|\boldsymbol{Lk}+\boldsymbol{v}|\ge\sqrt{R^2-\gamma}} e^{-\alpha|\boldsymbol{Lk}+\boldsymbol{v}|^2} \le \frac{d}{2}(\frac{2}{\rho})^d\Gamma(\frac{d}{2},(\sqrt{R^2-\gamma}-\frac{\rho}{2})^2), \quad (29)$$

where $\rho = \min\{\sqrt{\alpha}|\boldsymbol{Lk}| \mid \boldsymbol{k}\in\mathbb{Z}^d, \boldsymbol{k}\ne\boldsymbol{0}\}$. This completes the proof. $\square$

## C    FAST ALGORITHM OF POTENTIAL SUMMATION

### C.1    INTEGRAL TRANSFORMATION

We denote $G(\boldsymbol{L},\boldsymbol{v})$ as the potential summation and $U(\boldsymbol{L},\boldsymbol{v})$ as the potential function with a lattice matrix $\boldsymbol{L} \in \mathbb{R}^{d\times d}$ and a vector $\boldsymbol{v} \in \mathbb{R}^d$ between two atoms inside a unit cell. For a potential summation $S(\boldsymbol{a},\boldsymbol{b})$ of a crystal with lattice matrix $\boldsymbol{L}$, we have $S(\boldsymbol{a},\boldsymbol{b}) = G(\boldsymbol{L},\boldsymbol{v}_{ab})$. Based on these notations, we prove that the summation of the three introduced potentials can be transformed into an integral form as

$$G(\boldsymbol{L},\boldsymbol{v}) = D\sum_{\boldsymbol{k}\in\mathbb{Z}^d}\int_0^\infty t^{C-1}e^{-A\pi|\boldsymbol{Lk}+\boldsymbol{v}|^2t-\frac{B}{t}}dt, \quad (30)$$

where $A, B, C, D$ are constants derived from the corresponding specific potential forms.

*Proof.* We first prove that both the potential forms $U(\boldsymbol{L}, \boldsymbol{v}) = 1/|\boldsymbol{Lk} + \boldsymbol{v}|^{2p}$ and $U(\boldsymbol{L}, \boldsymbol{v}) = e^{-\alpha|\boldsymbol{Lk}+\boldsymbol{v}|}$ can be written in the following integral form

$$U(\boldsymbol{L}, \boldsymbol{v}) = D \int_0^\infty t^{C-1} e^{-A\pi|\boldsymbol{Lk}+\boldsymbol{v}|^2 t - \frac{B}{t}} dt. \tag{31}$$

1). For the potentials in the form of $U(\boldsymbol{L}, \boldsymbol{v}) = 1/|\boldsymbol{Lk} + \boldsymbol{v}|^{2p}$, we apply the Mellin transform such that

$$\begin{aligned} M\{U\}(\boldsymbol{L}, \boldsymbol{v}) &= \int_0^\infty t^{p-1} e^{-t|\boldsymbol{Ln}+\boldsymbol{v}|^2} dt \\ &= \frac{\Gamma(p)}{|\boldsymbol{Ln} + \boldsymbol{v}|^{2p}}. \end{aligned} \tag{32}$$

Thus, we obtain

$$U(\boldsymbol{L}, \boldsymbol{v}) = \frac{1}{|\boldsymbol{Lk} + \boldsymbol{v}|^{2p}} = \frac{1}{\Gamma(p)} \int_0^\infty t^{p-1} e^{-t|\boldsymbol{Ln}+\boldsymbol{v}|^2} dt. \tag{33}$$

Apparently, we can obtain $A = 1/\pi$, $B = 0$, $C = p$ and $D = 1/\Gamma(p)$ for the integral form of $U(\boldsymbol{L}, \boldsymbol{v}) = 1/|\boldsymbol{Lk} + \boldsymbol{v}|^{2p}$.

2). For the potentials in the form of $U(\boldsymbol{L}, \boldsymbol{v}) = e^{-\alpha|\boldsymbol{Lk}+\boldsymbol{v}|}$, we consider the inverse Laplace transform on $e^{-\alpha\sqrt{s}}$ as shown by Bateman (1954), such that

$$\mathcal{L}^{-1}\{e^{-\alpha\sqrt{s}}\} = \frac{a}{2\sqrt{\pi}} t^{-\frac{3}{2}} e^{-\frac{\alpha^2}{4t}}. \tag{34}$$

Therefore, we can apply the Laplace transform in Eqn. (34) to derive the integral form of $e^{-\alpha|\boldsymbol{Lk}+\boldsymbol{v}|}$:

$$\begin{aligned} U(\boldsymbol{L}, \boldsymbol{v}) &= e^{-\alpha|\boldsymbol{Lk}+\boldsymbol{v}|} \\ &= \frac{a}{2\sqrt{\pi}} \int_0^\infty t^{-\frac{3}{2}} e^{-|\boldsymbol{Lk}+\boldsymbol{v}|^2 t - \frac{\alpha^2}{4t}} dt \\ &= \frac{\alpha}{2\pi} \int_0^\infty t^{-\frac{3}{2}} e^{-\pi t|\boldsymbol{Lk}+\boldsymbol{v}|^2 - \frac{\alpha^2}{4\pi t}} dt \, (t \leftarrow \pi t). \end{aligned} \tag{35}$$

Apparently, we can obtain $A = 1$, $B = \frac{\alpha^2}{4\pi}$, $C = -\frac{1}{2}$ and $D = \frac{\alpha}{2\pi}$ for the integral form of $U(\boldsymbol{L}, \boldsymbol{v}) = e^{-\alpha|\boldsymbol{Lk}+\boldsymbol{v}|}$.

Finally, we conduct a summation for these two types of potentials $U(\boldsymbol{L}, \boldsymbol{v})$ in the space $\boldsymbol{k} \in \mathbb{Z}^d$ as

$$G(\boldsymbol{L}, \boldsymbol{v}) = \sum_{\boldsymbol{k} \in \mathbb{Z}^d} U(\boldsymbol{L}, \boldsymbol{v}). \tag{36}$$

This completes the proof. $\qquad\square$

In fact, the summation of $U(\boldsymbol{L}, \boldsymbol{v}) = 1/|\boldsymbol{Lk} + \boldsymbol{v}|^{2p}$ is a special case of multidimensional zeta function (Crandall & Buhler, 1987; Terras, 1973; Crandall, 1998), which is a generalization of Riemann zeta function. The multidimensional zeta function (Crandall & Buhler, 1987) is defined as

$$Z_{\boldsymbol{L}}(s; \boldsymbol{u}, \boldsymbol{v}) = \sum_{\boldsymbol{k} \in \mathbb{Z}^d} \frac{e^{2\pi i \boldsymbol{u} \cdot \boldsymbol{Lk}}}{|\boldsymbol{Lk} - \boldsymbol{v}|^s}, \tag{37}$$

where $s \in \mathbb{C}$, $\boldsymbol{L} \in \mathbb{R}^{d \times d}$ and $\boldsymbol{u}, \boldsymbol{v} \in \mathbb{R}^d$. It is not hard to show summation of $U(\boldsymbol{L}, \boldsymbol{v}) = 1/|\boldsymbol{Lk} + \boldsymbol{v}|^{2p}$ can be expressed as $Z_{\boldsymbol{L}}(2p; \boldsymbol{0}, -\boldsymbol{v})$, and therefore functional properties of the multidimensional zeta function also hold for summation of $U(\boldsymbol{L}, \boldsymbol{v}) = 1/|\boldsymbol{Lk} + \boldsymbol{v}|^{2p}$. For instance, $Z_{\boldsymbol{L}}(s; \boldsymbol{u}, \boldsymbol{v})$ has an analytic continuation to the entire complex plane, except for simple poles at $s = 0$ and $s = d$ (Crandall & Buhler, 1987). As a result, summation of $U(\boldsymbol{L}, \boldsymbol{v}) = 1/|\boldsymbol{Lk} + \boldsymbol{v}|^{2p}$ are all convergent for $p \in \mathbb{C}$ except $p = 0$ and $p = d/2$ by analytic continuation. Moreover, the multidimensional zeta function can be written in the form of an integral summation by Eqn. 33 as

$$Z_{\boldsymbol{L}}(s; \boldsymbol{u}, \boldsymbol{v}) = \sum_{\boldsymbol{k} \in \mathbb{Z}^d} \frac{e^{2\pi i \boldsymbol{u} \cdot \boldsymbol{Lk}}}{|\boldsymbol{Lk} - \boldsymbol{v}|^s} = \sum_{\boldsymbol{k} \in \mathbb{Z}^d} \frac{1}{\Gamma(s/2)} \int_0^\infty t^{s/2-1} e^{2\pi i \boldsymbol{u} \cdot \boldsymbol{Lk} - t|\boldsymbol{Lk} - \boldsymbol{v}|^2} dt. \tag{38}$$

Based on this, we can also split the integral and apply Poisson summation Eqn. 41 to obtain two summations of incomplete Bessel functions to evaluate this series. For more details on the multi-dimensional zeta function, we refer readers to Crandall & Buhler (1987); Terras (1973); Crandall (1998); Kirsten (1994); Selberg & Chowla (1967).

### C.2 Calculating Integral Summation

As shown in Sec. 3.4, $G(\boldsymbol{L}, \boldsymbol{v})$ can be written as the summations in the Euclidean space directly and then Fourier space:

$$
\begin{aligned}
G(\boldsymbol{L}, \boldsymbol{v}) &= D \sum_{\boldsymbol{k} \in \mathbb{Z}^d} \int_0^\infty t^{C-1} e^{-A\pi|\boldsymbol{Lk}+\boldsymbol{v}|^2 t - \frac{B}{t}} dt \\
&= D \sum_{\boldsymbol{k} \in \mathbb{Z}^d} \int_0^1 t^{C-1} e^{-A\pi|\boldsymbol{Lk}+\boldsymbol{v}|^2 t - \frac{B}{t}} dt + D \sum_{\boldsymbol{k} \in \mathbb{Z}^d} \int_1^\infty t^{C-1} e^{-A\pi|\boldsymbol{Lk}+\boldsymbol{v}|^2 t - \frac{B}{t}} dt \\
&= G_{\text{Fourier}}(\boldsymbol{L}, \boldsymbol{v}) + G_{\text{direct}}(\boldsymbol{L}, \boldsymbol{v}),
\end{aligned}
\tag{39}
$$

where $G_{\text{Fourier}}(\boldsymbol{L}, \boldsymbol{v}) = D \sum_{\boldsymbol{k} \in \mathbb{Z}^d} \int_0^1 t^{C-1} e^{-A\pi|\boldsymbol{Lk}+\boldsymbol{v}|^2 t - \frac{B}{t}} dt$ denotes the summation in Fourier space, and $G_{\text{direct}}(\boldsymbol{L}, \boldsymbol{v}) = D \sum_{\boldsymbol{k} \in \mathbb{Z}^d} \int_1^\infty t^{C-1} e^{-A\pi|\boldsymbol{Lk}+\boldsymbol{v}|^2 t - \frac{B}{t}} dt$ denotes the summation in direct space. Apparently, $G_{\text{direct}}(\boldsymbol{L}, \boldsymbol{v})$ is already the form of the incomplete Bessel function summation. We apply the analytic continuation to $G_{\text{direct}}(\boldsymbol{L}, \boldsymbol{v})$ to expand the domain of $C$ in $G_{\text{direct}}(\boldsymbol{L}, \boldsymbol{v})$ as detailed in Appendix C.3 and we have

$$
G_{\text{direct}}(\boldsymbol{L}, \boldsymbol{v}) = D \sum_{\boldsymbol{k} \in \mathbb{Z}^d} K_{-C}(A\pi|\boldsymbol{Lk}+\boldsymbol{v}|^2, B)
\tag{40}
$$

for all constant $C \in \mathbb{R}$. Below, we prove that $G_{\text{Fourier}}(\boldsymbol{L}, \boldsymbol{v})$ can be deduced into the incomplete Bessel function summation.

*Proof.* Inspired by the Ewald summation (Ewald, 1921; Crandall, 1998), we consider $G_{\text{Fourier}}(\boldsymbol{L}, \boldsymbol{v})$ on the reciprocal lattice using the Poisson summation (Crandall, 1998):

$$
\sum_{\boldsymbol{k} \in \mathbb{Z}^d} e^{2\pi i \boldsymbol{w} \cdot \boldsymbol{Lk} - \pi t |\boldsymbol{Lk}+\boldsymbol{v}|^2} = \frac{t^{-\frac{d}{2}} e^{2\pi i \boldsymbol{w} \cdot \boldsymbol{v}}}{\det \boldsymbol{L}} \sum_{\boldsymbol{k} \in \mathbb{Z}^d} e^{2\pi i \boldsymbol{L}' \boldsymbol{k} \cdot \boldsymbol{v} - \frac{\pi}{t}|\boldsymbol{L}'\boldsymbol{k}+\boldsymbol{w}|^2},
\tag{41}
$$

where $\boldsymbol{w} \in \mathbb{R}^d$ is a vector and $\boldsymbol{w} = \boldsymbol{0}$ in our case, and $\boldsymbol{L}' = \boldsymbol{L}(\boldsymbol{L}^T\boldsymbol{L})^{-1}$ is the lattice matrix for the reciprocal lattice. As a result, We obtain

$$
\begin{aligned}
G_{\text{Fourier}}(\boldsymbol{L}, \boldsymbol{v}) &= D \sum_{\boldsymbol{k} \in \mathbb{Z}^d} \int_0^1 t^{C-1} e^{-A\pi|\boldsymbol{Lk}+\boldsymbol{v}|^2 t - \frac{B}{t}} dt \\
&= \frac{D}{A^C} \sum_{\boldsymbol{k} \in \mathbb{Z}^d} \int_0^A t^{C-1} e^{-\pi|\boldsymbol{Lk}+\boldsymbol{v}|^2 t - \frac{AB}{t}} dt \, (t \leftarrow \frac{t}{A}) \\
&= \frac{1}{\det \boldsymbol{L}} \frac{D}{A^C} \sum_{\boldsymbol{k} \in \mathbb{Z}^d} \int_0^A t^{C-\frac{d}{2}-1} e^{2\pi i \boldsymbol{L}' \boldsymbol{n} \cdot \boldsymbol{v} - \frac{\pi}{t}|\boldsymbol{L}'\boldsymbol{k}|^2 - \frac{AB}{t}} dt \quad \text{(Eqn. (41))} \\
&= \frac{1}{\det \boldsymbol{L}} \frac{D}{A^{\frac{d}{2}}} \sum_{\boldsymbol{k} \in \mathbb{Z}^d} \int_0^1 t^{C-\frac{d}{2}-1} e^{2\pi i \boldsymbol{L}' \boldsymbol{k} \cdot \boldsymbol{v} - \frac{\pi}{At}|\boldsymbol{L}'\boldsymbol{k}|^2 - \frac{B}{t}} dt \, (t \leftarrow At) \\
&= \frac{1}{\det \boldsymbol{L}} \frac{D}{A^{\frac{d}{2}}} \sum_{\boldsymbol{k} \in \mathbb{Z}^d} \int_1^\infty t^{\frac{d}{2}-C-1} e^{2\pi i \boldsymbol{L}' \boldsymbol{n} \cdot \boldsymbol{v} - \frac{\pi t}{A}|\boldsymbol{L}'\boldsymbol{k}|^2 - Bt} dt \, (t \leftarrow \frac{1}{t}) \\
&= \frac{1}{\det \boldsymbol{L}} \frac{D}{A^{\frac{d}{2}}} \sum_{\boldsymbol{k} \in \mathbb{Z}^d} e^{2\pi i \boldsymbol{L}' \boldsymbol{k} \cdot \boldsymbol{v}} K_{C-\frac{d}{2}}(\frac{\pi|\boldsymbol{L}'\boldsymbol{k}|^2}{A} + B, 0).
\end{aligned}
\tag{42}
$$

We also apply the analytic continuation to $G_{\text{Fourier}}(\boldsymbol{L}, \boldsymbol{v})$ in the last step. Apparently, $G_{\text{Fourier}}(\boldsymbol{L}, \boldsymbol{v})$ is deduced into the incomplete Bessel function summation, and this completes the proof. $\qquad\square$

Therefore, both $G_{\text{Fourier}}(\boldsymbol{L}, \boldsymbol{v})$ and $G_{\text{direct}}(\boldsymbol{L}, \boldsymbol{v})$ can be expressed by the incomplete Bessel function summation forms:

$$
\begin{aligned}
G(\boldsymbol{L}, \boldsymbol{v}) &= G_{\text{direct}}(\boldsymbol{L}, \boldsymbol{v}) + G_{\text{Fourier}}(\boldsymbol{L}, \boldsymbol{v}) \\
&= D \sum_{\boldsymbol{k} \in \mathbb{Z}^d} K_{-C}(A\pi|\boldsymbol{L}\boldsymbol{k} + \boldsymbol{v}|^2, B) \\
&\quad + \frac{1}{\det \boldsymbol{L}} \frac{D}{A^{\frac{d}{2}}} \sum_{\boldsymbol{k} \in \mathbb{Z}^d} e^{2\pi i \boldsymbol{L}'\boldsymbol{k} \cdot \boldsymbol{v}} K_{C - \frac{d}{2}}\left(\frac{\pi|\boldsymbol{L}'\boldsymbol{k}|^2}{A} + B, 0\right).
\end{aligned}
\tag{43}
$$

As shown in Appendix B.2, the incomplete Bessel function summation $\sum_{\boldsymbol{k} \in \mathbb{Z}^d} K_\nu(\alpha|\boldsymbol{L}\boldsymbol{k} + \boldsymbol{v}|^2 + \gamma, \beta)$ is convergent and can be approximated. Therefore, $G(\boldsymbol{L}, \boldsymbol{v})$ can also be approximated.

## C.3 ANALYTIC CONTINUATION OF POTENTIAL SUMMATIONS

To represent the series that is not convergent, including the inverse summation $\sum_{\boldsymbol{k} \in \mathbb{Z}^d} \frac{1}{|\boldsymbol{L}\boldsymbol{k} + \boldsymbol{v}|}$ as shown in Sec. 3.4, we need to investigate the analytic continuation of potential summation. As shown in Appendix C.2, the potential summation can be written as

$$
\begin{aligned}
G(\boldsymbol{L}, \boldsymbol{v}) &= D \sum_{\boldsymbol{k} \in \mathbb{Z}^d} \int_0^\infty t^{C-1} e^{-A\pi|\boldsymbol{L}\boldsymbol{k} + \boldsymbol{v}|^2 t - \frac{B}{t}} dt \\
&= D \sum_{\boldsymbol{k} \in \mathbb{Z}^d} \int_0^1 t^{C-1} e^{-A\pi|\boldsymbol{L}\boldsymbol{k} + \boldsymbol{v}|^2 t - \frac{B}{t}} dt + D \sum_{\boldsymbol{k} \in \mathbb{Z}^d} \int_1^\infty t^{C-1} e^{-A\pi|\boldsymbol{L}\boldsymbol{k} + \boldsymbol{v}|^2 t - \frac{B}{t}} dt.
\end{aligned}
\tag{44}
$$

These two parts can be deduced into two incomplete Bessel function summations by analytic continuation. Analytic continuation is a technique to extend the domain $P$ of a given analytic function $f(x)$. Consequently, we denote $\hat{f}(x)$ as an analytic continuation of $f(x)$ to $Q$, where we denote the domain $Q$ containing $P$, and denote a function $\hat{f}(x)$ that is analytic on $Q$, and $\hat{f}(x) = f(x)$ holds for all $x$ in $P$. As shown by Kung & Yang (2003), the analytic continuation is unique and satisfies the permanence of functional relationships, i.e., the equations holding for $f(x)$ will also hold for $\hat{f}(x)$. In our case, we can expand the domain of constant $C$ in $G(\boldsymbol{L}, \boldsymbol{v})$ to $C \in \mathbb{R}$ such that $G(\boldsymbol{L}, \boldsymbol{v})$ is well-defined for any $C \in \mathbb{R}$. For example, for summation of the potentials in the form of $\sum_{\boldsymbol{k} \in \mathbb{Z}^d} 1/|\boldsymbol{L}\boldsymbol{k} + \boldsymbol{v}|^{2p}$, we have $C = p$ as shown in Appendix C.1. Analytic continuation enables us to compute the summation when $p = 0.5$, which is initially divergent. Formally, we assume that $C$ is originally defined in the domain $P \subset \mathbb{R}$. As derived in Appendix C.1, for $C \in P$, we have

$$
\begin{aligned}
G(\boldsymbol{L}, \boldsymbol{v}) &= D \sum_{\boldsymbol{k} \in \mathbb{Z}^d} \int_0^1 t^{C-1} e^{-A\pi|\boldsymbol{L}\boldsymbol{k} + \boldsymbol{v}|^2 t - \frac{B}{t}} dt + D \sum_{\boldsymbol{k} \in \mathbb{Z}^d} \int_1^\infty t^{C-1} e^{-A\pi|\boldsymbol{L}\boldsymbol{k} + \boldsymbol{v}|^2 t - \frac{B}{t}} dt \\
&= D \sum_{\boldsymbol{k} \in \mathbb{Z}^d} K_{-C}(A\pi|\boldsymbol{L}\boldsymbol{k} + \boldsymbol{v}|^2, B) \\
&\quad + \frac{1}{\det \boldsymbol{L}} \frac{D}{A^{\frac{d}{2}}} \sum_{\boldsymbol{k} \in \mathbb{Z}^d} e^{2\pi i \boldsymbol{L}'\boldsymbol{k} \cdot \boldsymbol{v}} K_{C - \frac{d}{2}}\left(\frac{\pi|\boldsymbol{L}'\boldsymbol{k}|^2}{A} + B, 0\right).
\end{aligned}
\tag{45}
$$

Furthermore, we have $\nu = -C$ and $\nu = C - \frac{d}{2}$ in the incomplete Bessel function, which is analytically continued to $\nu \in \mathbb{R}$ (Jones, 2007). That is, $P$ is contained by $\nu$'s domain $\mathbb{R}$. Therefore, the incomplete Bessel function summations are the analytic continuation of the corresponding potential summations.

Our approach is ultimately to capture the total contribution of the potential in the crystal system. Based on this, to explain the advantage of analytic continuation, we consider the total infinite potential summation inside a unit cell such that

$$
S = \sum_{\boldsymbol{a} \in \boldsymbol{A}} \sum_{\boldsymbol{b} \in \boldsymbol{A}} S(\boldsymbol{a}, \boldsymbol{b}) = \sum_{\boldsymbol{a} \in \boldsymbol{A}} \sum_{\boldsymbol{b} \neq \boldsymbol{a}, \boldsymbol{b} \in \widetilde{\boldsymbol{A}}} V(\boldsymbol{a}, \boldsymbol{b}).
\tag{46}
$$

where $\boldsymbol{A}$ and $\widetilde{\boldsymbol{A}}$ are described in Sec. 2.1. If we use an analytically continued function to approximate convergent summation $S(\boldsymbol{a}, \boldsymbol{b})$, we can directly approximate $S$ by

$$
S = \sum_{\boldsymbol{a} \in \boldsymbol{A}} \sum_{\boldsymbol{b} \in \boldsymbol{A}} S(\boldsymbol{a}, \boldsymbol{b}).
\tag{47}
$$

As for a non-convergent summation $\mathcal{S}(\boldsymbol{a}, \boldsymbol{b})$, due to the permanence of functional relationships of analytic continuation, we still obtain

$$S = \sum_{\boldsymbol{a} \in A} \sum_{\boldsymbol{b} \in A} S(\boldsymbol{a}, \boldsymbol{b}). \tag{48}$$

This implies that we can use analytically continued summations to approximate the total contribution of potentials where individual potential summations are initially divergent. In addition, for a non-convergent summation $\mathcal{S}(\boldsymbol{a}, \boldsymbol{b})$, analytic continuation will result in an unusual value such as negative energy instead of infinity. This is useful in practice since the infinity will cause a numerical explosion in training. A famous example of using analytic continuation for crystal energy prediction is the Madelung constant of $NaCl$, which is derived by the summation of Coulomb potentials among $Na$ and $Cl$ ions and calculated by analytic continuation of an absolutely convergent series (Borwein et al., 1985). We further show energy calculation of $NaCl$ by analytic continuation in Appendix E.

### C.4 IMPLEMENTATION AND NUMERICAL EXAMPLES OF APPROXIMATION

Here we describe the implementation of our algorithm as in Eqn. (43). Considering the fact that solving the inverse of the incomplete Gamma function is complicated, instead, we provide a proper value $R$ and then calculate its corresponding error bound $\epsilon$ based on Eqn. (29). Given a lattice matrix $\boldsymbol{L} \in \mathbb{R}^{d \times d}$, a vector $\boldsymbol{v} \in \mathbb{R}^d$ inside a unit cell, and the constants $A, B, C, D$ derived from specific potential functions as described in Appendix C.1, we aim to evaluate these two parts

$D \sum_{\boldsymbol{k} \in \mathbb{Z}^d} K_{-C}(A\pi|\boldsymbol{Lk} + \boldsymbol{v}|^2, B)$ and $\frac{1}{\det \boldsymbol{L}} \frac{D}{A^{\frac{d}{2}}} \sum_{\boldsymbol{k} \in \mathbb{Z}^d} e^{2\pi i \boldsymbol{L}'\boldsymbol{k} \cdot \boldsymbol{v}} K_{C - \frac{d}{2}}(\frac{\pi|\boldsymbol{L}'\boldsymbol{k}|^2}{A} + B, 0)$.

To evaluate $G_{\text{direct}}(\boldsymbol{L}, \boldsymbol{v}) = D \sum_{\boldsymbol{k} \in \mathbb{Z}^d} K_{-C}(A\pi|\boldsymbol{Lk} + \boldsymbol{v}|^2, B)$, we derive the following steps.

**Step 1:** Determine the value $R$ such that $R^2 > C$, $R^2 > 1$, and calculate the error bound $\epsilon = \frac{d}{2}(\frac{2}{\rho})^d \Gamma(\frac{d}{2}, (R - \frac{\rho}{2})^2)$, where $\rho = \min\{\sqrt{A\pi}|\boldsymbol{Lk}| \,|\, \boldsymbol{k} \in \mathbb{Z}^d, \boldsymbol{k} \neq \boldsymbol{0}\}$.

**Step 2:** Select points inside an ellipsoid such that $\boldsymbol{P} = \{\boldsymbol{k}| \,|\boldsymbol{Lk} + \boldsymbol{v}| \leq R/\sqrt{A\pi}\}$.

**Step 3:** Evaluate the incomplete Bessel function summation $G_{\text{direct}}(\boldsymbol{L}, \boldsymbol{v})$ by calculating the every term $DK_{-C}(A\pi|\boldsymbol{Lk} + \boldsymbol{v}|^2, B)$ based on Appendix B.1.

To evaluate $G_{\text{Fourier}}(\boldsymbol{L}, \boldsymbol{v}) = \frac{1}{\det \boldsymbol{L}} \frac{D}{A^{\frac{d}{2}}} \sum_{\boldsymbol{k} \in \mathbb{Z}^d} e^{2\pi i \boldsymbol{L}'\boldsymbol{k} \cdot \boldsymbol{v}} K_{C - \frac{d}{2}}(\frac{\pi|\boldsymbol{L}'\boldsymbol{k}|^2}{A} + B, 0)$, we derive the following steps.

**Step 1:** Determine the value $R$ such that $R^2 > \frac{d}{2} - C$, $R^2 > 1$, $R^2 > B$, and calculate the error bound $\epsilon > \frac{d}{2}(\frac{2}{\rho})^d \Gamma(\frac{d}{2}, (\sqrt{R^2 - B} - \frac{\rho}{2})^2)$, where $\rho = \min\{\sqrt{\frac{\pi}{A}}|\boldsymbol{L}'\boldsymbol{k}| \,|\, \boldsymbol{k} \in \mathbb{Z}^d, \boldsymbol{k} \neq \boldsymbol{0}\}$.

**Step 2:** Select points inside an ellipsoid such that $\boldsymbol{P} = \{\boldsymbol{k}| \,|\boldsymbol{Lk} + \boldsymbol{v}| \leq \sqrt{A(R^2 - B)/\pi}\}$.

**Step 3:** Evaluate the incomplete Bessel function summation $G_{\text{Fourier}}(\boldsymbol{L}, \boldsymbol{v})$ by calculating every term $\frac{1}{\det \boldsymbol{L}} \frac{D}{A^{\frac{d}{2}}} e^{2\pi i \boldsymbol{L}'\boldsymbol{k} \cdot \boldsymbol{v}} K_{C - \frac{d}{2}}(\frac{\pi|\boldsymbol{L}'\boldsymbol{k}|^2}{A} + B, 0)$ based on Appendix B.1.

Our implementation is based on Cython and GNU Scientific Library (Galassi et al., 2002), in which the native incomplete Gamma function and Bessel function are used to implement the incomplete Bessel function. We conduct numerical experiments on Intel Xeon Gold 6258R CPU. We show the evaluation examples in Table 6 with the corresponding error bound and evaluation time. The running time is at the scale of milliseconds.

### C.5 POTENTIAL SUMMATION EXTENSIONS

To highlight the generality of our potential summation method, in this section, we introduce additional potentials that can be converted to our general integral form in Eqn. 31, including Lennard-Jones potential, Morse potential, and screened Coulomb potential. These potentials are only used for some specific types of materials, such as gas and fluid materials.

**Lennard-Jones Potential** (Lennard-Jones & Dent, 1928) is an intermolecular pair potential that is usually used for gas or organic materials. The commonly used expression for the Lennard-Jones

Table 6: Numerical examples of our algorithm. Here, $\zeta(x) = \sum_{n=1}^{\infty} 1/n^x$. Implementation details can be found in Appendix C.4.

| Ground Truth | Evaluation | Estimated Error | Real Error | Time |
|---|---|---|---|---|
| $2 * \zeta(2) = 3.28986813$ | 3.28068288 | 8e-1 | 9e-3 | 0.002 s |
| $2 * \zeta(2) = 3.28986813$ | 3.28984070 | 6e-2 | 3e-5 | 0.003 s |
| $2 * \zeta(2) = 3.28986813$ | 3.28986812 | 7e-4 | 1e-8 | 0.003 s |
| $2 * \zeta(2) = 3.28986813$ | 3.28986813 | 1e-6 | < 1e-8 | 0.003 s |
| $2 * \zeta(3) = 2.40411381$ | 2.40411381 | 1e-6 | < 1e-8 | 0.003 s |
| $2 * \zeta(4) = 2.16464647$ | 2.16464647 | 1e-6 | < 1e-8 | 0.003 s |
| $\sum_{n \in \mathbb{Z}^2, n \neq 0} \frac{1}{\|n\|^4} = 6.02681204$ | 5.99068949 | 3 | 4e-2 | 0.003 s |
| $\sum_{n \in \mathbb{Z}^2, n \neq 0} \frac{1}{\|n\|^4} = 6.02681204$ | 6.02670959 | 2e-1 | 1e-4 | 0.003 s |
| $\sum_{n \in \mathbb{Z}^2, n \neq 0} \frac{1}{\|n\|^4} = 6.02681204$ | 6.02681199 | 3e-3 | 5e-8 | 0.003 s |
| $\sum_{n \in \mathbb{Z}^2, n \neq 0} \frac{1}{\|n\|^4} = 6.02681204$ | 6.02681204 | 5e-6 | < 1e-8 | 0.003 s |
| $\sum_{n \in \mathbb{Z}} e^{-\|n\|} = 2.16395341$ | 2.16395326 | 4e-1 | 2e-7 | 0.002 s |
| $\sum_{n \in \mathbb{Z}} e^{-\|n\|} = 2.16395341$ | 2.16395341 | 3e-4 | < 1e-8 | 0.004 s |
| $\sum_{n \in \mathbb{Z}^3} e^{-\|n\|} = 25.39268269$ | 25.39268214 | 2.5 | 5e-7 | 0.003 s |
| $\sum_{n \in \mathbb{Z}^3} e^{-\|n\|} = 25.39268269$ | 25.39268269 | 1e-2 | < 1e-8 | 0.003 s |

potential is

$$U_{LJ}(\boldsymbol{L}, \boldsymbol{v}) = 4\epsilon \left[ \left( \frac{\sigma}{|\boldsymbol{L}\boldsymbol{k} + \boldsymbol{v}|} \right)^{12} - \left( \frac{\sigma}{|\boldsymbol{L}\boldsymbol{k} + \boldsymbol{v}|} \right)^{6} \right], \tag{49}$$

where $\epsilon$ and $\sigma$ are hyperparameters. And the summation of $U_{LJ}(\boldsymbol{L}, \boldsymbol{v})$ can be converted to two potential summations of type $1/|\boldsymbol{L}\boldsymbol{k} + \boldsymbol{v}|^{2p}$ with $p = 3$ and $p = 6$, such that

$$G_{LJ}(\boldsymbol{L}, \boldsymbol{v}) = \sum_{\boldsymbol{k} \in \mathbb{Z}^d} U_{LJ}(\boldsymbol{L}, \boldsymbol{v}) = 4\epsilon \left[ \sigma^{12} \left( \sum_{\boldsymbol{k} \in \mathbb{Z}^d} \frac{1}{|\boldsymbol{L}\boldsymbol{k} + \boldsymbol{v}|^{12}} \right) - \sigma^6 \left( \sum_{\boldsymbol{k} \in \mathbb{Z}^d} \frac{1}{|\boldsymbol{L}\boldsymbol{k} + \boldsymbol{v}|^6} \right) \right], \tag{50}$$

where we show the calculation of the potential summation of type $1/|\boldsymbol{L}\boldsymbol{k} + \boldsymbol{v}|^{2p}$ in Appendix C.1.

**Morse Potential** (Morse, 1929) is an interatomic potential of a diatomic molecule and can be used for simple molecular materials. The Morse potential has a mathematical form of

$$U_{Morse}(\boldsymbol{L}, \boldsymbol{v}) = D_e \left( e^{-2a(|\boldsymbol{L}\boldsymbol{k}+\boldsymbol{v}|-r_e)} - 2e^{-a(|\boldsymbol{L}\boldsymbol{k}+\boldsymbol{v}|-r_e)} \right), \tag{51}$$

where $D_e$ and $r_e$ are hyperparameters. Similarly, the summation of $U_{Morse}(\boldsymbol{L}, \boldsymbol{v})$ can be converted to two potential summations of type $e^{-\alpha|\boldsymbol{L}\boldsymbol{k}+\boldsymbol{v}|}$ with $\alpha = a$ and $\alpha = 2a$, such that

$$G_{Morse}(\boldsymbol{L}, \boldsymbol{v}) = \sum_{\boldsymbol{k} \in \mathbb{Z}^d} U_{Morse}(\boldsymbol{L}, \boldsymbol{v}) = D_e \left( e^{2ar_e} \sum_{\boldsymbol{k} \in \mathbb{Z}^d} e^{-2a|\boldsymbol{L}\boldsymbol{k}+\boldsymbol{v}|} - 2e^{ar_e} \sum_{\boldsymbol{k} \in \mathbb{Z}^d} e^{-a|\boldsymbol{L}\boldsymbol{k}+\boldsymbol{v}|} \right), \tag{52}$$

where we show the calculation of the potential summation of type $e^{-\alpha|\boldsymbol{L}\boldsymbol{k}+\boldsymbol{v}|}$ in Appendix C.1.

**Screened Coulomb Potential** or Yukawa potential (Yukawa, 1935) represents the Coulomb interactions with damping of electric fields. It is an important potential reflecting the behaviors of charge-carrying fluids or particles in semiconductors. The screened Coulomb potential has an analytic form of $V(\boldsymbol{a}, \boldsymbol{b}) = \frac{z_{\boldsymbol{a}} z_{\boldsymbol{b}} e^2}{d(\boldsymbol{a}, \boldsymbol{b})} \exp(-\alpha d(\boldsymbol{a}, \boldsymbol{b}))$ where $d(\boldsymbol{a}, \boldsymbol{b})$ is the distance between atom $\boldsymbol{a}$ and $\boldsymbol{b}$, $z_{\boldsymbol{a}}, z_{\boldsymbol{b}}$ are charges of atom $\boldsymbol{a}$ and $\boldsymbol{b}$, $e$ is elementary charge constant and $\alpha$ is a scaling hyperparameter. Since $z_{\boldsymbol{a}}, z_{\boldsymbol{b}}, e$ are constants and can be extracted outside the summation, we can obtain the simplified screened Coulomb potential

$$U_{screened}(\boldsymbol{L}, \boldsymbol{v}) = \frac{e^{-\alpha|\boldsymbol{L}\boldsymbol{k}+\boldsymbol{v}|}}{|\boldsymbol{L}\boldsymbol{k} + \boldsymbol{v}|}. \tag{53}$$

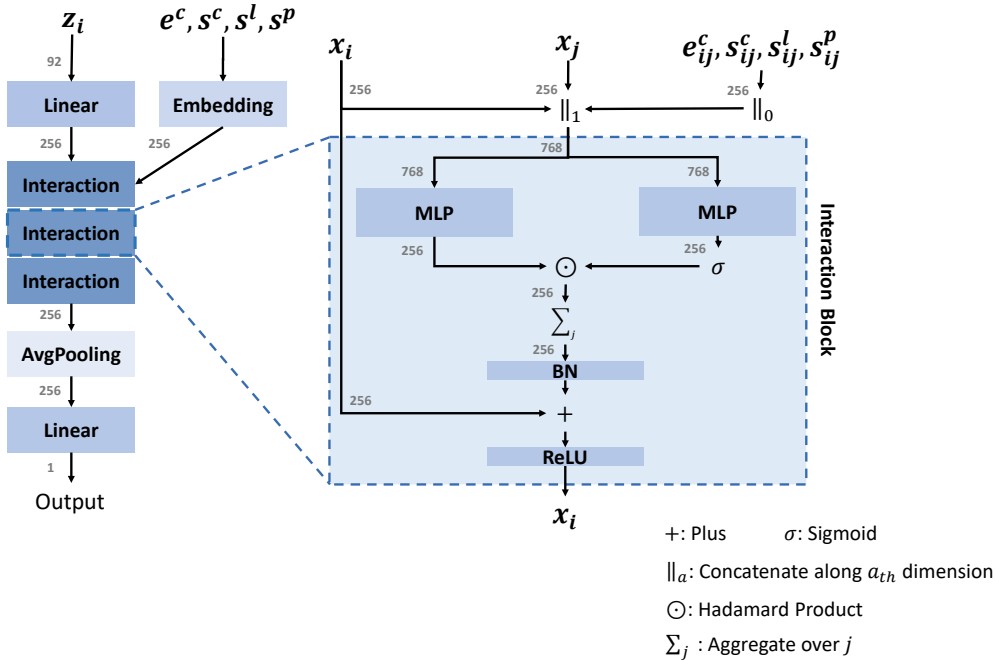

Figure 2: The developed network architecture for PotNet.

Consider the inverse Laplace transform on $e^{-\alpha\sqrt{s}}/\sqrt{s}$, by using Bromwich contour with branch points, we obtain

$$\mathcal{L}^{-1}\{e^{-\alpha\sqrt{s}}/\sqrt{s}\} = \frac{1}{\sqrt{\pi t}}e^{-\frac{\alpha^2}{4t}}. \tag{54}$$

Therefore, we can apply the Laplace transform in Eqn. 53 such that

$$U_{screened}(\boldsymbol{L}, \boldsymbol{v}) = \frac{e^{-\alpha|\boldsymbol{Lk}+\boldsymbol{v}|}}{|\boldsymbol{Lk}+\boldsymbol{v}|} = \frac{1}{\sqrt{\pi}}\int_0^\infty t^{-\frac{1}{2}}e^{-|\boldsymbol{Lk}+\boldsymbol{v}|^2 t - \frac{\alpha^2}{4t}}\,dt. \tag{55}$$

Then we can obtain $A = 1/\pi$, $B = \alpha^2$, $C = \frac{1}{2}$ and $D = 1/\sqrt{\pi}$ in Eqn. 30 to fit screened Coulomb potential into our potential summation method.

## D  MODEL IMPLEMENTATION

### D.1  POTNET IMPLEMENTATION

The employed network architecture is shown in Fig. 2. Since our major contribution is to consider infinite interatomic potentials, we simply design our network architecture following the commonly used settings. Specifically, existing methods for 3D graphs (Xie & Grossman, 2018; Schütt et al., 2017; Klicpera et al., 2020b;a; Liu et al., 2022b; Gasteiger et al., 2021; Schütt et al., 2021) share a similar architecture, which usually contains an input block, an interaction block, and an output block. Without loss of generality, we take the updating process for node $i$ as an example to illustrate the network.

*The Inputs* contain atomic features and potentials. $\mathbf{z}_i$ is the 92-dimensional atomic feature for any atom $i$ following CGCNN (Xie & Grossman, 2018). Below we denote $d$ as the interatomic distances and show our potential features. For our implementation of infinite potential summations $S(\boldsymbol{a}, \boldsymbol{b})$ in Eqn. (7), we add up the infinite summations of three potentials, including the Coulomb potentials, London dispersion potentials, and Pauli repulsion potentials described in Sec. 2.2. We simplify the mathematical form of Coulomb potentials $V_{\text{Coulomb}}(\boldsymbol{a}, \boldsymbol{b}) = -\frac{z_a z_b e^2}{4\pi\epsilon_0 d(\boldsymbol{a}, \boldsymbol{b})}$ to $V_{\text{Coulomb}}(\boldsymbol{a}, \boldsymbol{b}) = -\epsilon_1/d(\boldsymbol{a}, \boldsymbol{b})$ where $\epsilon_1$ is a hyperparameter, because $e, \pi, \epsilon_0$ are all known constants, and $z_a, z_b$ can

be learned from atomic features. As explained in Sec. 4.1, $\mathbf{e}^c = -\epsilon'_1/d$ denotes the Coulomb potentials for the local crystal graph. $\mathbf{s}^c = -\sum_d \epsilon_1/d, \mathbf{s}^l = -\sum_d \epsilon/d^6, \mathbf{s}^p = \sum_d e^{-\alpha d}$ denote the summations of Coulomb potentials, London dispersion potentials, and Pauli potentials for the infinite crystal graph, and we set $\epsilon'_1 = 4.0$ for $\mathbf{e}^c$, $\epsilon_1 = -1.0$ for $\mathbf{s}^c$, $\epsilon = 4.0$ for $\mathbf{s}^l$ and $\alpha = 3.0$ for $\mathbf{s}^p$, respectively. For simplicity, we apply them to the RBF embeddings with the same cutoff. Since $\mathbf{e}^c$ and $\mathbf{s}^l$ are negative values and have values exceeding the RBF cutoff, we use an exponential function to make their absolute values smaller.

*The input block* contains a ***Linear*** layer and an ***Embedding*** layer. For each node $i$, the ***Linear*** layer is employed to generate a 256-dimensional vector as the input node features to the first interaction layer. For each edge, the ***Embedding*** layer is employed to map the Coulomb potentials and summations of Coulomb potentials, London dispersion potentials, and Pauli repulsion potentials to 256-dimensional embeddings by using 256 RBF kernels with centers from -4.0 to 4.0.

*The Interaction block* contains several ***Interaction*** layers. Each layer updates the feature vector of node $i$ based on features of the neighboring nodes and potential embeddings of the connected edges. Particularly, for any of the neighboring node $j$ of node $i$, the corresponding potential embeddings $\mathbf{e}^c_{ij}, \mathbf{s}^c_{ij}, \mathbf{s}^l_{ij}$, and $\mathbf{s}^p_{ij}$ for edge $ij$ are all produced by the ***Embedding*** layer.

*The Readout block* contains an ***AvgPooling*** layer and another ***Linear*** layer. We first use the ***AvgPooling*** layer to aggregate features from all nodes in a graph and then use the ***Linear*** layer that maps the hidden dimension of 256 to the final output which is a scalar.

## D.2 CONFIGURATIONS OF RETRAINED MODELS

In this section, we show detailed configurations of retrained models of CGCNN (Xie & Grossman, 2018), SchNet (Schütt et al., 2017), MEGNET (Chen et al., 2019), GATGNN (Louis et al., 2020) and ALIGNN (Choudhary & DeCost, 2021). If not specified, models are trained with a radius cutoff of 8.0 using the Adam (Kingma & Ba, 2014) optimizer with weight decay (Loshchilov & Hutter, 2017) and one cycle learning rate scheduler (Smith & Topin, 2019).

**CGCNN** (Xie & Grossman, 2018). We directly use the publicly available code from Xie & Grossman (2018) to build and train the CGCNN model. We build the model with 128 hidden dimensions and three message-passing layers and train the model for 1000 epochs with a batch size of 256 and an initial learning rate of 1e-2.

**SchNet** (Schütt et al., 2017). We directly adopt the SchNet model from PyTorch Geometric (Fey & Lenssen, 2019) with 128 hidden dimensions and six message-passing layers following the original paper. We train SchNet with an initial learning rate of 5e-4 and batch size of 64 for 500 epochs.

**GATGNN** (Louis et al., 2020). We directly use the publicly available code from Louis et al. (2020) to build and train the GATGNN model. We build the model with 128 hidden dimensions and three message-passing layers following the original default settings. We train the model with an initial learning rate of 5e-3 and batch size of 64 for 500 epochs.

**MEGNET** (Chen et al., 2019). We directly adopt the MEGNET model from the publicly available code from Chen et al. (2019). Following the original paper, we use three message-passing layers with the same feature dimensions as mentioned in the original paper and use Set2Set readout function. We train MEGNET with an initial learning rate of 1e-3 and batch size of 128 for 1000 epochs following the configuration settings mentioned in the original paper.

**ALIGNN** (Choudhary & DeCost, 2021). We directly use the publicly available code from Choudhary & DeCost (2021). We use the official best model configurations of ALIGNN to train ALIGNN models with an initial learning rate of 1e-3 and batch size of 64 for 500 epochs.

## E LINEAR ENERGY MODELING USING INFINITE POTENTIAL SUMMATION

In this section, we provide examples of calculations where the true energies of these materials can be directly approximated by linear combinations of our infinite potential sums. These are special cases of Eqn. 6 with a linear embedded function $G$. Specifically, we evaluate the total energy per atom of $NaCl$ and two other materials ($MgO$, $LiF$) whose crystal structures are similar to $NaCl$.

Since they are pure ionic crystals and Coulomb interactions dominate the system, we first consider the total electrostatic energy and Coulomb potential summations. For a neutral system, the electrostatic energy is convergent because the Coulomb potentials cancel each other. To calculate the energy of a crystal, we cannot derive it directly due to its complexity. Instead, we divide it into many individual infinite potential summations. Although these individual summations might be originally divergent, they will all be well-defined under analytic continuation and the total energy will also be convergent.

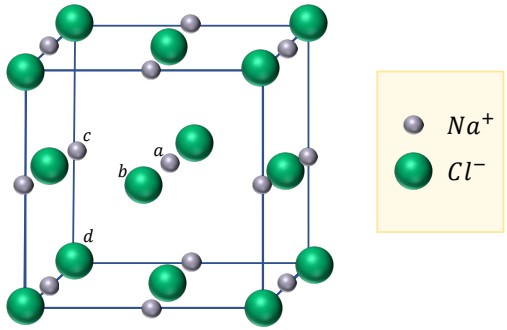

Figure 3: Crystal structure of $NaCl$.

As shown in Appendix C.3, analytic continuation allows us to compute potential summations that are initially divergent, and also allows us to compute the convergent summations by analytically continued potential summations. For example, we can calculate $\sum_{n\in\mathbb{Z},n\neq 0}\frac{(-1)^n}{n}$ by analytic continuation such that

$$\sum_{n\in\mathbb{Z},n\neq 0}\frac{(-1)^n}{n} = \sum_{n=1}^{\infty}\frac{1}{n} - \sum_{n=0}^{\infty}\frac{1}{n+\frac{1}{2}} = \zeta(1,1) - \zeta(1,\frac{1}{2}),\tag{56}$$

where $\zeta(s,a)$ is Hurwitz zeta function and $\zeta(s,a) = \sum_{n=0}^{\infty}\frac{1}{(n+a)^s}$ when $s > 1, a \neq 0, -1, -2, ...$, and its analytic continuation elsewhere. That is, we can use the analytically continued zeta function $\zeta(s,a)$ to precisely approximate a convergent series.

Inspired by analytic continuation, we can approximate the total energy per atom of crystals of $NaCl$ by analytically continued infinite potential summations. Due to the symmetry of the $NaCl$ cell, we only involve atoms $a, b, c, d$ in our calculation. Specifically, as shown in Fig. 3, atom $a$ represents the body center $Na^+$, atom $b$ represents the face center $Cl^-$, atom $c$ represents the edge center $Na^+$ and atom $d$ represents the corner $Cl^-$. Based on this, assuming that the side length of the unit cell is 1, the total energy of $Na^+$ is approximated by the total Coulomb interactions with atom $a$ such that

$$\begin{aligned}
E_{Na} &= -N_A\left[1\cdot\frac{z_{Na}z_{Na}e^2}{4\pi\epsilon_0}\sum_{\boldsymbol{u}\in\boldsymbol{A_a},\boldsymbol{u}\neq\boldsymbol{a}}\frac{1}{d(\boldsymbol{a},\boldsymbol{u})} + \frac{1}{2}6\cdot\frac{z_{Na}z_{Cl}e^2}{4\pi\epsilon_0}\sum_{\boldsymbol{u}\in\boldsymbol{A_b}}\frac{1}{d(\boldsymbol{a},\boldsymbol{u})}\right.\\
&\quad \left.+ \frac{1}{4}12\cdot\frac{z_{Na}z_{Na}e^2}{4\pi\epsilon_0}\sum_{\boldsymbol{u}\in\boldsymbol{A_c}}\frac{1}{d(\boldsymbol{a},\boldsymbol{u})} + \frac{1}{8}8\cdot\frac{z_{Na}z_{Cl}e^2}{4\pi\epsilon_0}\sum_{\boldsymbol{u}\in\boldsymbol{A_d}}\frac{1}{d(\boldsymbol{a},\boldsymbol{u})}\right]\\
&= -N_A\left[\frac{z_{Na}z_{Na}e^2}{4\pi\epsilon_0 r_0}\sum_{\boldsymbol{u}\in\boldsymbol{A_a},\boldsymbol{u}\neq\boldsymbol{a}}\frac{1}{\tilde{d}(\boldsymbol{a},\boldsymbol{u})} + 3\cdot\frac{z_{Na}z_{Cl}e^2}{4\pi\epsilon_0 r_0}\sum_{\boldsymbol{u}\in\boldsymbol{A_b}}\frac{1}{\tilde{d}(\boldsymbol{a},\boldsymbol{u})}\right.\\
&\quad \left.+ 3\cdot\frac{z_{Na}z_{Na}e^2}{4\pi\epsilon_0 r_0}\sum_{\boldsymbol{u}\in\boldsymbol{A_c}}\frac{1}{\tilde{d}(\boldsymbol{a},\boldsymbol{u})} + \frac{z_{Na}z_{Cl}e^2}{4\pi\epsilon_0 r_0}\sum_{\boldsymbol{u}\in\boldsymbol{A_d}}\frac{1}{\tilde{d}(\boldsymbol{a},\boldsymbol{u})}\right]\\
&= -\frac{N_A|z_{Na}||z_{Cl}|e^2}{4\pi\epsilon_0 r_0}\left[\tilde{\mathcal{S}}(\boldsymbol{a},\boldsymbol{a}) - 3\cdot\tilde{\mathcal{S}}(\boldsymbol{a},\boldsymbol{b}) + 3\cdot\tilde{\mathcal{S}}(\boldsymbol{a},\boldsymbol{c}) - \tilde{\mathcal{S}}(\boldsymbol{a},\boldsymbol{d})\right]\\
&\approx -\frac{N_A|z_{Na}||z_{Cl}|e^2}{4\pi\epsilon_0 r_0}\left[-1.41864874 - 3\cdot(-0.04796615) + 3\cdot(-0.29126077)\right.\\
&\quad \left.- (-0.40096799)\right]\\
&\approx -\frac{N_A|z_{Na}||z_{Cl}|e^2}{4\pi\epsilon_0 r_0}\cdot(-1.7475646)\\
&\approx 8.81eV,
\end{aligned}\tag{57}$$

where $N_A$ is Avogadro constant, $d(\boldsymbol{a},\boldsymbol{u})$ is the distance between atom $\boldsymbol{a}$ and $\boldsymbol{u}$, $r_0$ is the minimum distance between $Na$ and $Cl$, $\tilde{d} = d/r_0$ is the normalized distance, $\boldsymbol{A_a}$ denotes the set of atoms containing atom $\boldsymbol{a}$ and all its repetitions, $\tilde{\mathcal{S}}$ is the infinite potential summation, approximated by

our infinite potential summation method in Sec. 3.4, $z_{Na}, z_{Cl}$ are charges of $Na^+$ and $Cl^-$, $e$ is the elementary charge constant, $\epsilon_0$ is the permittivity constant of free space, and the coefficients $1, \frac{1}{2}6, \frac{1}{4}12, \frac{1}{8}8$ denote the fraction of atoms in a unit cell. We finally obtain a constant $-1.7475646$ from our infinite potential summations. In fact, this constant is exactly the famous Madelung constant $M$ (Borwein et al., 1985). Also, by considering an additional repulsion term, we can derive the calculation result in Eqn. 57 to the famous Born-Landé equation (Born, 1921)

$$E = -\frac{N_A|z^+||z^-|e^2 M}{4\pi\epsilon_0 r_0}(1 - \frac{1}{n}), \tag{58}$$

where $z^+, z^-$ are the charges of cation and anion, $M$ is the Madelung constant computed from infinite Coulomb potential summations, and $n$ is the Born exponent measuring the effect of repulsion. By choosing $n = 9$, we can derive an approximation for the total energy of $NaCl$ of 7.84 eV. Similarly, we can apply Eqn. 58 to $MgO$ and

Table 7: Total energy per atom approximation of $NaCl$, $MgO$ and $LiF$.

| Formula | $r_0$ | $n$ | Ground Truth | Eqn. 58 |
|---------|-------|-----|--------------|---------|
| $NaCl$ | 282 pm | 9 | 8.15 eV | 7.84 eV |
| $MgO$ | 210 pm | 6 | 39.33 eV | 39.45 eV |
| $LiF$ | 201 pm | 7 | 10.67 eV | 10.60 eV |

$LiF$. We further show these approximations in Table. 7 and it can be noticed that these approximations already give rough results compared to the ground truth energy. This implies that our features can serve as a good starting point for machine learning models to learn the ground truth energy. Apparently, previous methods cannot achieve this due to the lack of such informative features. It is worth noting that the Madelung constant is typically unknown because those coefficients for the infinite potential summations depend on the charge distribution in the system, which we do not know at the beginning. Also, these crystals are special cases of Eqn. 6 with a linear embedded function $G$, while $G$ is typically a nonlinear function (Daw & Baskes, 1984). Therefore, the network serves the purpose of learning those coefficients to learn the Madelung constant and providing nonlinearity.

