# OpenReview forum: "Efficient Approximations of Complete Interatomic Potentials for Crystal Property Prediction"
_ICLR.cc/2023/Conference — Submitted to ICLR 2023_

### Official Review · Reviewer_FZse · 2022-10-20

**Confidence:** 5
**Correctness:** 1
**Technical Novelty And Significance:** 3
**Empirical Novelty And Significance:** Not applicable
**Recommendation:** 3

**Clarity, Quality, Novelty And Reproducibility:**

The paper is written clearly in good quality. The idea of the paper is novel. It is not reproducible due to the wrong math.

**Strength And Weaknesses:**

Strength: the motivation and idea of the paper is awesome. Traditional neural networks for molecular property prediction usually use local information as atom features, which limits the description for long-range interactions. However, unlike short range interactions which involves complicated electronic structure, the formula of long-range interactions are usually easy. So incorporate fix-formula is a promising solution for long-range interactions.

Weakness: The paper has a fatal mathematical error, which invalidate the computation of Coulomb part. The error occurs in Appendix C.1. In formula (31), it is ... e^{-|Lk+v|t}, but in formula (32), it is miswrote as ... e^{-t|Lk+v|^2}. In appendix C.2, C.3 and C.4, it is e^{-t|Lk+v|^2} everywhere. This error is fatal because if we change e^{-t|Lk+v|^2} back to the correct e^{-t|Lk+v|}, the Poisson summation (39) does not hold, so the summation of Fourier term (40) also does not hold. In fact, if the authors did not make the mistake, the analytic continuation of infinite summation of Coulomb potentials should be diverged, similarly to the fact that Riemann zeta function diverges at s=1.

**Summary Of The Paper:**

This paper proposed to incorporate Coulomb potential, London dispersion potential and Pauli repulsion potential in the input features of deep learning models, in order to deal with the long range interactions in PBC systems like crystal materials. It is proved that the proposed model outperforms SOTA in two material datasets: The Materials Project and JARVIS.

**Summary Of The Review:**

This motivation for using closed-form formula to describe long-range interactions in neural network is a great idea. However, a fatal mathematical error (see above) make the computation of Coulomb term completely wrong, which diminish the contribution of this paper.

---

> ### Author Response · Authors · 2022-11-08
> **Responses to Reviewer FZse**
>
> We genuinely appreciate your review. We sincerely apologize that we have typos in our paper. The "errors" you mentioned are actually typos in our paper. Below we provide two detailed responses to address the weakness stated above. We also revised the paper accordingly.
>
> > **Q: The paper has a fatal mathematical error, which invalidate the computation of Coulomb part. The error occurs in Appendix C.1. In formula (31), it is ... $e^{-|Lk+v|t}$, but in formula (32), it is miswrote as ... $e^{-t|Lk+v|^2}$. In appendix C.2, C.3 and C.4, it is $e^{-t|Lk+v|^2}$ everywhere. This error is fatal because if we change $e^{-t|Lk+v|^2}$ back to the correct $e^{-t|Lk+v|}$, the Poisson summation (39) does not hold, so the summation of Fourier term (40) also does not hold.**
>
> - We sincerely apologize for these typos. Specifically, we miss the squared symbols in Sec 3.4 and Appendix C.1. In formula (30) and (31), it should be $e^{-t|Lk+v|^{\color{red}2}}$ rather than $e^{-t|Lk+v|}$. Therefore, the formula (30) should be written as
>
> $$
> G(L, v) = D \int_0^\infty t^{C-1} (-\delta(v,B) + \sum_{k \in \mathbb{Z}^d}e^{-A\pi |Lk+v|^2 t - \frac{B}{t}}) dt,
> $$
>
> - As you may notice, for the same formula, *we do have the squared symbols in the other parts, like Appendix C.2 and C.3 (for example, you can easily find it in formula (37) and formula (42) in Appendix C.2 and C.3).* However, we miss the squared symbols at formula (8) in Sec 3.4 and formula (30) in Appendix C.1. Also, in Appendix C.1, the Coulomb potential type should be $\sum_{k\in \mathbb{Z}^3,|Lk+v|\ne 0} 1/|Lk+v|^{{\color{red}2}p}$ where there is a squared symbol on the top. The Pauli potential type proof in the formula (35) should also have a squared symbol like $e^{-t|Lk+v|^{\color{red}2}}$. We sincerely apologize for these typos. And these typos occurred when we moved the formulas of Sec 3.4 and Appendix C.1 from hand-written papers to the electronic version. We highly appreciate you pointing this out and we edit Sec 3.4 and Appendix C.1 in ${\color{red} red }$. After editing, Sec 3.4 and Appendix C.1 should be consistent with all other sections.
> - Since they are typos, they won’t affect **all other theoretical proofs and all experimental results** of our methods. All of our theoretical proofs are based on $|Lk+v|^2$ in the integral in the sections discussed above. Also, all of our experimental implementations are also based on $|Lk+v|^2$ and experimental results will remain the same in Sec 4 and Appendix C.4.
>
> > **Q: If the authors did not make the mistake, the analytic continuation of infinite summation of Coulomb potentials should be diverged, similarly to the fact that Riemann zeta function diverges at $s=1$.**
>
> - Thank you for raising this. **The analytic continuation of infinite summation of Coulomb potentials converges.** We list the reasons below:
>
>   - The Riemann zeta function does diverge at $s=1$. However, we should note that this happens since it's on the **one-dimensional lattice**. Our series is computed on the **three-dimensional lattice**. In our case, the respective zeta function is a three-dimensional zeta function $Z(s) = \sum_{k\in \mathbb{Z}^3, |Lk+v|\ne 0} 1/|Lk+v|^s$. And the pole should be $s=3$ rather than $s=1$. Generally, the $d$-dimensional zeta function $Z(s) = \sum_{k\in \mathbb{Z}^d, |Lk+v|\ne 0} 1/|Lk+v|^s$ has two poles $s=0$ and $s=d$, i.e., $Z(s)$ diverges only when $s=0$ and $s=d$. We refer this to \[1\],\[2\],\[3\] and \[4\].
>   - The analytical continuation of Coulomb potential summation is computed on the three-dimensional lattice with $s=1$ and $d=3$. Therefore, it should converge.
>
> - We think the relation with Riemann zeta function is a good point, so we revised our paper and added discussions at the end of Appendix C.1 in ${\color{blue} blue }$.
>
> **We appreciate it if you could re-evaluate our work based on the revised paper and our responses. If you have any further concerns, we are more than glad to answer.**
>
> References:
>
> \[1\] Crandall, R. E., & Buhler, J. P. (1987). Elementary function expansions for Madelung constants. Journal of Physics A: Mathematical and General, 20(16), 5497.
>
> \[2\] Terras, A. A. (1973). Bessel series expansions of the Epstein zeta function and the functional equation. Transactions of the American Mathematical Society, 183, 477-486.
>
> \[3\] Kirsten, K. (1994). Generalized multidimensional Epstein zeta functions. Journal of Mathematical Physics, 35(1), 459-470.
>
> \[4\] Epstein zeta-function. Encyclopedia of Mathematics. URL: http://encyclopediaofmath.org/index.php?title=Epstein_zeta-function&oldid=42020

---

> > ### Comment · Reviewer_FZse · 2022-12-12
> > **I'm still not clear**
> >
> > Sorry for the late reply, but I think I'm still not clear. Let me try to explain step by step. All the formula numbers refer to the renewed paper.
> >
> > - Starting from Coulomb term 1/|Lk+v|.
> > - In formula (33), you want to change 1/|Lk+v| into the integral form, so 2p=1, C=p=1/2. A = 1/pi and B=0.
> > - In formula (42), you transform the integral into reciprocal space, where in the last line it will be K_{C-d/2} (pi |L'k|^2/A+B, 0). Since A=1/pi, B=0, C=1/2, it will be K_{-1/2-d/2}(pi^2 |L'k|^2, 0).
> > - Now since you take all summation over k in Z^d, let's talk about k = (0, 0, ..., 0) first. In this case it will be K_{-1/2-d/2}(0,0)
> > - Go back to the definition of incomplete Bessel function in (13), K_{-1/2-d/2}(0,0) = \int_1^{\infty} t^{d/2-1/2} dt. For any d >=1, this integration is diverged.
> >
> > Do you think my arguments are correct, or is there any ways to fix this?

---

> ### Author Response · Authors · 2022-12-11
> **A gentle reminder to Reviewer FZse**
>
> Dear Reviewer FZse,
>
> Thanks again for your valuable comments! We posted our rebuttal on November 8th. We have provided substantial explanations and also revised the paper heavily to address your concerns. Could you please reply to our rebuttal and let us know if there are any additional concerns? As this is the only negative review and it is approaching the end of the discussion period, we eagerly look forward to receiving your feedback. Thank you so much!
>
> Sincerely,
>
> Authors

---

### Official Review · Reviewer_q5TT · 2022-10-21

**Confidence:** 3
**Correctness:** 3
**Technical Novelty And Significance:** 3
**Empirical Novelty And Significance:** 3
**Recommendation:** 6

**Clarity, Quality, Novelty And Reproducibility:**

The paper is fairly clear, and (at first read to me appeared) technically sound. The main novelty of the paper is the application of Ewald summation within a GNN.


**Strength And Weaknesses:**


Strengths:
Great example of how incorporation of physical knowledge into a GNN structure leads to a more efficient architecture.

The methods is demonstrated on several examples, and ablation reveals the importance of the infinite summation to handle long range interactions.

Weaknesses:

The description of Ewald summation in the main paper is very concise - it would be nice with a bit more intuition about the methods for readers that are not familiar with this. Also, it is not clear to what extent this is just application of a known result/method, or if this is a novel mathematical result. In case there is something substantially new in the methods, this should be in the main paper in my opinion. I did not carefully check the derivations in the appendix.

I expected to see comparison with other GNN approaches where the cutoff was varied, to be able to more directly understand the importance of the infinite summation. It is not clear from the paper (but from the references) what the cutoff was set to in e.g. the SchNet comparison (5Å i think)

EDIT: (after reading FZse's review) Since the electrostatic (Coulomb) potential is summed infinitely for each (charged) atom, does that not mean that these individual sums in general will diverge, even though the system is electrically neutral?


**Summary Of The Paper:**

The paper proposes a graph neural network (GNN) for property prediction for crystals. A key issue in modeling crystals is to handle long range interactions. This is handled by using interatomic potentials as edge features in the GNN and approximating the infinite summation over all pairwise interactions using Ewald summation.


**Summary Of The Review:**

Good paper with a simple but interesting and novel idea. Reasonable empirical evaluation and comparison. The point raised by FZse should be addressed.

---

> ### Author Response · Authors · 2022-11-14
> **Responses to Reviewer q5TT (Part III)**
>
> > **Q: Since the electrostatic (Coulomb) potential is summed infinitely for each (charged) atom, does that not mean that these individual sums in general will diverge, even though the system is electrically neutral?**
>
> - We apologize for the mathematical typos in Sec. 3.4 and Appendix C.1 as we explain for reviewer FZse. For this question, it will not diverge for these individual summations due to the analytical continuation as we already discussed in Sec. 3.4 and Appendix C.3. We also present the intuitive explanation here.
>
> - Analytic continuation is a mathematical technique used to extend the domain of a given function, for example, by defining a new analytic region in which an infinite sequence is initially divergent. By using this technique, we can prevent infinity in the calculation, but also maintain the properties of the function in the original domain \[12\].
>
> - For a neutral system, the energy is convergent because the Coulomb potentials cancel each other. To calculate the energy of a crystal, we cannot derive it directly due to its complexity. Instead, we divide it into many individual infinite potential summations. Although these individual summations might be originally divergent, they will all be well-defined under analytical continuation and the total energy will also be convergent \[10\]\[11\]. This approach is very similar to the following example of using analytical continuation to calculate $\sum_{n\in\mathbb{Z}, n\ne 0} \frac{(-1)^n}{\sqrt{n}}$ such that
> \begin{equation}
> \sum_{n\in\mathbb{Z}, n\ne 0} \frac{(-1)^n}{\sqrt{n}}
> = \sum_{n=0}^\infty \frac{2}{\sqrt{2n + 2}} -  \sum_{n=0}^\infty \frac{2}{\sqrt{2n + 1}}
> = \sqrt{2}\left(\zeta(\frac{1}{2}, 1) - \zeta(\frac{1}{2}, \frac{1}{2})\right),
> \end{equation}
> where $\zeta(s, a)$ is Hurwitz zeta function and $\zeta(s, a) = \sum_{n=0}^\infty \frac{1}{(n+a)^s}$ when $s > 1, a\ne 0,-1,-2,...$, and its analytic continuation elsewhere. That is, we can use the analytically continued zeta function $\zeta(s, a)$ to precisely approximate a convergent series. This equation holds due to the permanence of functional relationships as shown in [12], i.e., the functional properties holding for $f(x)$ will also
> hold for its analytic continuation $\hat{f}(x)$.
>
> - To further explain this, we provide some calculation examples to approximate the energy of materials by this technique as shown in Appendix E. We can observe that by a linear combination of Coulomb potential summations via analytical continuation, we can calculate the total Coulomb contribution of the system and approximate the energy of these materials.
>
> \[1\] Woodward, Philipp M. (1953). Probability and Information Theory, with Applications to Radar. Academic Press, p. 36.
>
> \[2\] Lee, H., & Cai, W. (2009). Ewald summation for Coulomb interactions in a periodic supercell. Lecture Notes, Stanford University, 3(1), 1-12.
>
> \[3\] Nestler, F., Pippig, M., & Potts, D. (2015). Fast Ewald summation based on NFFT with mixed periodicity. Journal of Computational Physics, 285, 280-315.
>
> \[4\] Crandall, R. E. (1998). Fast evaluation of Epstein zeta functions.
>
> \[5\] Schütt, K. T., Sauceda, H. E., Kindermans, P. J., Tkatchenko, A., & Müller, K. R. (2018). Schnet–a deep learning architecture for molecules and materials. The Journal of Chemical Physics, 148(24), 241722.
>
> \[6\] Choudhary, K., & DeCost, B. (2021). Atomistic Line Graph Neural Network for improved materials property predictions. npj Computational Materials, 7(1), 1-8.
>
> \[7\] Xie, T., & Grossman, J. C. (2018). Crystal graph convolutional neural networks for an accurate and interpretable prediction of material properties. Physical review letters, 120(14), 145301.
>
> \[8\] Chen, C., Ye, W., Zuo, Y., Zheng, C., & Ong, S. P. (2019). Graph networks as a universal machine learning framework for molecules and crystals. Chemistry of Materials, 31(9), 3564-3572.
>
> \[9\] Louis, S. Y., Zhao, Y., Nasiri, A., Wang, X., Song, Y., Liu, F., & Hu, J. (2020). Graph convolutional neural networks with global attention for improved materials property prediction. Physical Chemistry Chemical Physics, 22(32), 18141-18148.
>
> \[10\] Crandall, R. E., & Buhler, J. P. (1987). Elementary function expansions for Madelung constants. Journal of Physics A: Mathematical and General, 20(16), 5497.
>
> \[11\] Borwein, D., Borwein, J. M., & Taylor, K. F. (1985). Convergence of lattice sums and Madelung’s constant. Journal of mathematical physics, 26(11), 2999-3009.
>
> \[12\] Kung, Joseph PS & Yang, Chung-Chun. (2003). Complex Analysis. Encyclopedia of Physical Science and Technology.

---

> > ### Comment · Reviewer_q5TT · 2022-11-15
> > **Thank you for the detailed response**
> >
> > I appreciate your effort to address the potential issues I have pointed out, and I think the paper has been substantially improved in the new revision. I can now see how the issue with divergence of the individual sums is handled with the analytic continuation (although I have still not carefully checked the details). I have adjusted my score.

---

> ### Author Response · Authors · 2022-11-14
> **Responses to Reviewer q5TT (Part II)**
>
> > **Q: Also, it is not clear to what extent this is just application of a known result/method, or if this is a novel mathematical result. In case there is something substantially new in the methods, this should be in the main paper in my opinion. I did not carefully check the derivations in the appendix.**
>
> - Our potential summation method is both novel and general compared to previous methods. We add a small paragraph to highlight our novelty of our potential summation method in Sec. 3.4 in ${\color{blue}{blue}}$. We also add a new section Appendix C.5 to show that our potential summation method is general to most of interatomic potentials.
>
> - To our best knowledge, our method is novel since we are the first to use the incomplete Bessel function to compute the potential summations. By doing this, our method is able to compute a more general form of potential summation. Specifically, we can compute both the Pauli potential summation $\sum_{k\in\mathbb{Z}^d} e^{-\alpha |Lk + v|}$ and Coulomb potential summation $\sum_{k\in\mathbb{Z}^d} 1/{|Lk+v|}^s$. However, the previous methods such as \[2\]\[3\]\[4\] use Ewald summation for potential summation computation, which can only work for Coulomb potential summation $\sum_{k\in\mathbb{Z}^d} 1/{|Lk+v|}^s$.
>
> - We also want to point out that our method is general and is able to compute most of interatomic potentials. Besides the potentials we use in the main paper, we are also able to compute Lennard-Jones potential, Morse potential, and screened Coulomb potential by using our method as shown in Appendix C.5. These potentials are important for some specific materials, such as organic or fluid materials. The generality of the computation also benefits from the general form of our potential summation method. We show detailed derivation on Lennard-Jones potential, Morse potential, and screened Coulomb potential by our method respectively in Appendix C.5.
>
> > **Q: I expected to see comparison with other GNN approaches where the cutoff was varied, to be able to more directly understand the importance of the infinite summation. It is not clear from the paper (but from the references) what the cutoff was set to in e.g. the SchNet comparison (5Å i think)**
>
> - Thanks for the valuable suggestions. We have reproduced all other previous approaches by a cutoff of 8Å as we follow the previous SOTA method ALIGNN \[6\]. To provide more information about our reproduced method settings, we also add a new section Appendix D.2 to discuss all of our implementation settings of SchNet \[5\], CGCNN \[7\], MegNet \[8\], GATGNN \[9\] and ALIGNN \[6\] in detail.
>
> - We choose two of the approaches to evaluate with varied cutoffs. One is SchNet \[5\] as you comment and the other one is GATGNN \[9\] which is different from other methods since it applies global attention. We perform training and testing on these two approaches in JARVIS dataset and report the results w.r.t formation energy as below.
>
> | Cutoff | SchNet | GATGNN|
> | ------ | ------ | ----- |
> | 4      |   0.052     |  0.048     |
> | 8      |   0.045     |  0.047     |
> | 12     |   0.045     |   0.047    |
> | 16     |   0.045     |   0.046    |
> | 20     |   0.044     |   0.046    |
> | 30     |   0.042     |   0.045    |
> | 50     |   Unstable     |   0.043   |
>
> - As shown in the above table, we can observe a tendency that as the cutoff becomes larger, the performance becomes better. However, when the cutoff is as large as 50Å, the training of SchNet becomes unstable and the final result explodes, which we think is due to the different capacity of models. We also note that as the cutoff becomes larger, more time will be consumed for preprocessing. We show the preprocessing time with different cutoffs in the following table, and we can notice that the preprocessing time becomes more and more intolerant as the cutoff becomes larger.
>
>
> | Cutoff | Preprocessing Time |
> | ------ | ------  |
> | 4      | 84 s    |
> | 8      | 108 s   |
> | 12     | 112 s   |
> | 16     | 178 s   |
> | 20     | 259 s   |
> | 30     | 688 s   |
> | 50     | 3239 s  |

---

> ### Author Response · Authors · 2022-11-14
> **Responses to Reviewer q5TT (Part I)**
>
> Thank you for your review. We appreciate your insightful feedback and concerns. We integrate all your helpful feedback into the paper and respond to your concerns as outlined below.
>
> > **Q: The description of Ewald summation in the main paper is very concise - it would be nice with a bit more intuition about the methods for readers that are not familiar with this.**
>
> - Thank you for raising this issue. We add a small paragraph to explain Ewald summation in Sec. 3.4 in ${\color{blue}{blue}}$ and also explain the intuition here.
>
> - Typically, to evaluate a discrete convergent summation $\sum_{n=-\infty}^{\infty} f(n)$, we can set a cutoff $R$ around a center $C$ such that truncation error of the summation $\sum_{n\in \mathbb{Z}, |n - C| > R } f(n)$ is bounded. However, in many cases, the summation converges slowly and truncation error cannot be expressed as a closed-form function of $R$, such as $\sum_{n=-\infty, n\ne 0}^\infty 1/n^2$.
>
> - Fortunately, by using Fourier transform, we can transform this summation into another summation in the Fourier space. In this Fourier space, the lengths of the basis are proportional to the inverse ones in the original real space. As a result, the points that are loosely scattered in the real space will become tighter, and vice versa, resulting in different convergence rates of the summations in these two spaces. Actually, a slowly converging summation in real space is guaranteed to be converted to an equivalent but quickly converging summation in Fourier space \[1\].
>
> - Hence, in our paper, we adopt an effective strategy (Ewald summation) that we divide a discrete convergent summation into two parts. One part has a quicker converging rate in real space than the original summation. The other "slower-to-converge" part is transformed into the Fourier space and the convergence speed becomes quick. Then we can efficiently evaluate the original summation by adding up these two quickly convergent summations.

---

### Official Review · Reviewer_FEhN · 2022-10-24

**Confidence:** 4
**Correctness:** 3
**Technical Novelty And Significance:** 3
**Empirical Novelty And Significance:** 3
**Recommendation:** 6

**Clarity, Quality, Novelty And Reproducibility:**

The paper is well-written, and the description is detailed and clear.

There are some minor problems in the paper: the units of the compared physical quantities are not clarified throughout the paper (e.g. Table 1 and Table 2); the importance of the infinite potential summations is not assessed.

The novelty is good. It has two main innovations in the field of crystal material modeling using graph neutral networks. It is the first time that the interatomic potentials are directly as edge features and it is the first time to models the complete set of potentials among all atoms though there are some approximations.

Considering the detailed description was given in the paper, the proposed method is reproducible, but it is still complicated in practice.

**Strength And Weaknesses:**

Strength:

The proposed method improves the accuracy of the GNN-based interatomic potentials for crystal property prediction.

Weaknesses:

1. Compared with ALIGNN method, PotNet method consumes about four times as much time when infinite potential summations are considered, but the gain in accuracy is not that obvious.

2. The comparisons in efficiency between the proposed method and direct DFT calculations was not made. Similarly, the advantages and disadvantages of the proposed method compared with the machine learning potentials using the constructed vectors as crystal structure input.

**Summary Of The Paper:**

This paper proposes a new graph deep learning method, PotNet, with several innovations in the field of crystal material modeling. The proposed method models interatomic potentials directly as edge features instead of only using distances and models the complete set of potentials among all atoms with approximations of infinite potential summations instead of only between nearby atoms. With the computations of complete interatomic potentials, the representation learning is performed through message passing neural networks. Evaluation experiments are also performed on the JARVIS and Materials Project benchmarks, showing the proposed method leads to consistent performance improvements with reasonable computational costs.

**Summary Of The Review:**

This paper proposed a novel graph deep learning method, PotNet, which can improve the performance of the graph neural networks potentials in the field of crystal material modeling with certain computational costs. However, compared with the existing method, the gain of the proposed method in accuracy is not remarkable with the time costs of about 4 times. It is better to illuminate the specific application situations where the proposed method has more pronounced effect or further improve the algorithms to achieve faster implementation.

---

> ### Author Response · Authors · 2022-11-14
> **Responses to Reviewer FEhN**
>
> Thank you for your time and comments. We understand that you have a major concern about the efficiency of our method. We clarify that the time cost of our method is reasonable and the overall efficiency of our method is higher than ALIGNN and significantly higher than the DFT method.
>
> >  **Q: Compared with ALIGNN method, PotNet method consumes about four times as much time when infinite potential summations are considered, but the gain in accuracy is not that obvious. The comparisons in efficiency between the proposed method and direct DFT calculations was not made.**
> - The time you stated here is actually the **preprocessing time** of the whole JARVIS dataset as shown in Table 4. It is not included in the training or inference procedure of PotNet. To perform preprocessing, unlike previous methods like ALIGNN, we need to compute the complete potential set besides constructing graphs. Therefore, we cannot avoid spending more time to compute. However, for a single material, the preprocessing time of our method is at the level of milliseconds. Even we additionally compute the infinite potential summations, the preprocessing time of our method and ALIGNN is within the same order of magnitude. Also, compared to DFT method, the preprocessing time of our method is totally acceptable since DFT usually takes time from minutes to hours to process a single material. Overall, the computational cost of our method is reasonable.
> - We would like to highlight that for the training and inference time, our PotNet model surpasses ALIGNN by a large margin as shown in Table 3. For example, in training, PotNet takes 48 seconds per epoch while ALIGNN takes 327 seconds per epoch.
> - To further demonstrate efficiency of PotNet, we provide the overall inference time (**considering both preprocessing time and model inference time**) as shown in the following table. Specifically, we compare our method to ALIGNN and a DFT method. We observe that PotNet has a faster inference speed than ALIGNN. Also, PotNet surpasses the DFT method w.r.t speed by orders of magnitude.
>
> | Inference Time| PotNet | ALIGNN | DFT(K-Point Sampling)|
> | ---------------------- | ------ | ------ | ----- |
> | Single Material | 18 ms  | 30 ms| $\approx$ 10 min |
> | JARVIS Testing Set   | 101 s | 171 s |  > 1 day |
>
> - Especially, for conducting DFT experiments, we use k-point sampling via Vienna Ab initio Simulation Package (VASP) in high-throughput clusters. We observe that for a structure with approximately 10 atoms in a cell, using DFT from scratch on 20-40 cores will take 10 minutes (20 seconds every iteration for 30 to 50 steps) on average, which is at a different time scale compared to our method. Also, we note that for JARVIS dataset the mean number of atoms in a cell is 10. As a result, we cannot finish the processing of DFT in JARVIS testing set in one day.
> - We totally admit the confusion of Table 4. To avoid this, we edit the content of Table 4 and revise the related texts in Sec. 4.2 in ${\color{blue}{blue}}$.
>
> >  **Q: Similarly, the advantages and disadvantages of the proposed method compared with the machine learning potentials using the constructed vectors as crystal structure input.**
> - Sorry but we do not quite understand the question. Could you please provide more details about what you want us to compare with, or provide more references?
>
> > **Q: The units of the compared physical quantities are not clarified throughout the paper (e.g. Table 1 and Table 2);**
> - Thanks for pointing that out. We have added all the physical quantities in Table 1 and Table 2.
>
> > **Q: The importance of infinite potential summations is not assessed.**
> - Theoretically, the importance of our infinite potential summation has been discussed at length in Sec. 3.1 and Sec. 3.2. Concretely, in Sec. 3.1, we show that the energy of a crystal can be directly modeled by our infinite potential summation with an embedded function as shown in formula (3). This indicates that the energy is highly correlated with our infinite potential summations. **In the second bullet of Sec. 3.2, we show that missing the infinite potential summations will lead to a significant error.**
> - Experimentally, we have assessed the importance of our infinite potential summation by ablation studies experiments in Sec. 4.3. Specifically, by adding our infinite potential summations, we can **observe the performance gains and obtain the state-of-the-art result** as shown in Table 5.
> - Additionally, to further highlight the importance of the infinite potential summation, we add a new section Appendix E in ${\color{blue}{blue}}$. We show that for some crystals (like NaCl, MgO, etc), **their true energies can be directly approximately by a linear combination of our infinite potential summation**. This implies that our features can serve as a good starting point for machine learning models to learn the ground truth energy. Apparently, previous methods cannot achieve this due to the lack of such informative features.

---

> > ### Comment · Reviewer_FEhN · 2022-12-10
> > **Thank you for the thorough response**
> >
> > I really appreciate your effort giving a response to my review. I have carefully read it as well as the other reviews and responses. I think this work contains some contributions that deserve to be published, so I've updated the score.

---

### Author Response · Authors · 2022-11-17
**To Reviewer FEhN and Reviewer FZse**

Dear Reviewers,

Thank you for your reviews. Our point-by-point responses can be found below. We also have revised our paper heavily based on your suggestions and comments. Could you please take a look at the response and the revision, and let us know if we have addressed all your concerns? We are also happy to continue discussions if you still have remaining concerns.

Thanks for your time!

Sincerely,

Authors

---

### Author Response · Authors · 2022-12-06
**A gentle reminder to Reviewer FEhN and FZse**

Dear Reviewer FEhN and FZse,

We posted our detailed response and made heavy revisions to our paper three weeks ago. As the discussion period is coming to an end, could you please check our responses and revisions and let us know if we have addressed your concerns properly? If you still have additional concerns, we would be very happy to continue the discussion.

Thanks for your time!

Sincerely,

Authors

---

### Comment · Area_Chair_GiMe · 2022-12-07
**Response to Author Feedback**

Dear Reviewers, thank you so much again for your time on this paper. Thank you also Reviewer q5TT for responding to the author's feedback. Other reviewers: the discussion phase is still ongoing, how does the author response and other reviews change your view of the paper?

---

### Decision · Program_Chairs · 2023-01-20

**Decision:**

Reject

**Justification For Why Not Higher Score:**

See above metareview.

**Justification For Why Not Lower Score:**

N/A

**Metareview: Summary, Strengths And Weaknesses:**

The reviewers were split about this paper and did not come to a consensus: on one hand they appreciated the importance of the problem addressed and the clarity of the writing, on the other they had doubts about some of the derivations in the paper. After going through the paper and the discussion I have decided to vote to reject for the following reason: the main methodological idea of the paper is to describe a way to formulate the energy of a crystal that takes into account interatomic interactions and potentials in a principled way. However, this leads to infinite sums, so they describe an efficient approximation algorithm to estimate this based on (Ewald, 1921). Unfortunately, the original paper has a number of errors in the derivations, as pointed out by Reviewer FZse. The authors respond that these are typos that do not affect other theoretical proofs or any experimental results, and describe fixes. Reviewer FZse replies that these fixes are not sufficient to resolve the issues. The Reviewer agree and describe more fixes and point out again that this doesn’t affect their experimental results because it is based on the algorithm of Crandall et al., 1998. The Reviewer and authors respond two more times in this way: the reviewer pointing out a derivation issue with the paper/response, and the authors arguing either that (a) they have made a typo in the paper, (b) they misinterpreted the Reviewer’s response, or (c) they actually use the correct formula in another paper, all of which do not change their experimental results. By this point I have lost faith in the author’s ability to write a correct derivation, to clearly communicate any confusion about the derivation, and to write code that avoided all of these issues. The reviewers (particularly FZse) have given extremely detailed feedback and I recommend the authors follow their suggestions closely (if not already done) before submitting to a future ML venue. If the authors are able to fix these things it will make a much stronger submission.